# Damage sensing mediated by serine proteases Hayan and Persephone for Toll pathway activation in apoptosis-deficient flies

Shotaro Nakano[1], Soshiro Kashio[1], Kei Nishimura[1], Asuka Takeishi[2], Hina Kosakamoto[3], Fumiaki Obata [3,4], Erina Kuranaga[5], Takahiro Chihara[6], Yoshio Yamauchi[7], Toshiaki Isobe[7], Masayuki Miura[1] *

1 Department of Genetics, Graduate School of Pharmaceutical Sciences, The University of Tokyo, Tokyo, Japan, 2 Neural Circuit of Multisensory Integration RIKEN Hakubi Research Team, RIKEN Center for Brain Science, RIKEN Cluster for Pioneering Research, Wako, Japan, 3 Laboratory for Nutritional Biology, RIKEN Center for Biosystems Dynamics Research, Kobe, Japan, 4 Laboratory of Molecular Cell Biology and Development, Graduate School of Biostudies, Kyoto University, Kyoto, Japan, 5 Laboratory of Histogenetic Dynamics, Graduate School of Life Sciences, Tohoku University, Sendai, Japan, 6 Program of Biomedical Science and Program of Basic Biology, Graduate School of Integrated Sciences for Life, Hiroshima University, Higashi-Hiroshima, Hiroshima Japan, 7 Department of Chemistry, Graduate School of Science, Tokyo Metropolitan University, Hachioji, Japan

* miura@mol.f.u-tokyo.ac.jp

**Data Availability Statement:** All relevant data other than RNA-seq analysis and Proteome analysis are within the manuscript and its Supporting

## Abstract

The mechanisms by which the innate immune system senses damage have been extensively explored in multicellular organisms. In *Drosophila*, various types of tissue damage, including epidermal injury, tumor formation, cell competition, and apoptosis deficiency, induce sterile activation of the Toll pathway, a process that requires the use of extracellular serine protease (SP) cascades. Upon infection, the SP Spätzle (Spz)-processing enzyme (SPE) cleaves and activates the Toll ligand Spz downstream of two paralogous SPs, Hayan and Persephone (Psh). However, upon tissue damage, it is not fully understood which SPs establish Spz activation cascades nor what damage-associated molecules can activate SPs. In this study, using newly generated uncleavable *spz* mutant flies, we revealed that Spz cleavage is required for the sterile activation of the Toll pathway, which is induced by apoptosis-deficient damage of wing epidermal cells in adult *Drosophila*. Proteomic analysis of hemolymph, followed by experiments with *Drosophila* Schneider 2 (S2) cells, revealed that among hemolymph SPs, both SPE and Melanization Protease 1 (MP1) have high capacities to cleave Spz. Additionally, in S2 cells, MP1 acts downstream of Hayan and Psh in a similar manner to SPE. Using genetic analysis, we found that the upstream SPs Hayan and Psh contributes to the sterile activation of the Toll pathway. While *SPE/MP1* double mutants show more impairment of Toll activation upon infection than *SPE* single mutants, Toll activation is not eliminated in these apoptosis-deficient flies. This suggests that Hayan and Psh sense necrotic damage, inducing Spz cleavage by SPs other than SPE and MP1. Furthermore, hydrogen peroxide, a representative damage-associated molecule, activates the Psh-Spz cascade in S2 cells overexpressing Psh. Considering that reactive oxygen species (ROS) were detected in apoptosis-deficient wings, our findings highlight the importance

Information files. RNA-seq data are deposited in the DNA Data Bank of Japan (DDBJ) under accession no. DRA014922. Proteome data are deposited in the ProteomeXchange Consortium via the jPOST partner repository with the dataset identifier PXD037066.

**Funding:** This work was supported by grants from the Ministry of Education, Culture, Sports, Science, and Technology of Japan (KAKENHI Grant Numbers 16H06385, 21H04774 and 23H04766 to M.M). The funders had no role in study design, data collection and analysis, decision to publish, or preparation of the manuscript.

**Competing interests:** The authors have declared that no competing interests exist.

of ROS as signaling molecules that induce the activation of SPs such as Psh in response to damage.

## Author summary

The innate immune system is conserved among multicellular organisms and has developed the mechanisms to recognize various dangers including infection and non-infectious damage. In *Drosophila*, the Toll pathway, an innate immune signaling pathway, is activated in response to tissue damage. Toll activation requires the use of serine protease (SP) cascades to activate the Toll ligand Spätzle. In this study, we explored SPs and damage-associated molecules involved in the sterile activation of the Toll pathway using a fly model, where apoptosis deficiency of wing epidermal cells causes damage. Genetic analyses demonstrated that paralogous SPs Hayan and Persephone contribute to Toll activation in apoptosis-deficient flies. Furthermore, we found that hydrogen peroxide, a representative damage-associated molecule, activates Persephone in cells and that apoptosis-deficient wings produce reactive oxygen species (ROS). Our findings highlight the importance of ROS in initiating the SP cascade for Toll activation in response to tissue damage.

## Introduction

The mechanisms by which the innate immune system of insects senses damage has not yet been clearly explored. In contrast, in mammals, pattern recognition receptors (PRRs) such as Toll-like receptors (TLRs) can directly recognize molecules that originate from the site of injury, termed damage-associated molecular patterns (DAMPs) [1,2]. In *Drosophila*, the Toll pathway, a major immune signaling pathway, is activated in response to tissue damage caused by clean injury, tumor formation, brain aging, cell competition, or apoptosis deficiency [3–13]. The physiological functions of the Toll pathway in such contexts have been reported in detail. Upon clean injury to the larval or adult epidermis, Toll signaling is activated in fat bodies, circulating hemocytes, and the larval lymph gland (hematopoietic organ), exerting immune responses through expression of effector proteins, such as antimicrobial peptides (AMPs), and hemocyte differentiation to protect the organism against subsequent infection and parasitization [5,6]. In tumor-bearing larvae, one of the AMPs, defensin, is secreted downstream of the Toll pathway, and suppresses tumor growth by promoting cell death [8]. Although our understanding of the functions of the Toll pathway has advanced, the molecular mechanisms by which the Toll pathway is activated upon damage have been less explored.

Amplification of signals by proteolytic cascades allows for highly sensitive detection and response to infection. Such signal amplification was originally described in the hemolymph clotting system of infected horseshoe crabs (*Tachypleus tridentatus*) [14,15], in which CLIP-domain serine proteases were involved in this cascade. Subsequently, it was shown that this cascade also plays an important role in the *Drosophila* Toll pathway [16]. Toll receptors are activated by interacting with the protein ligand Spätzle (Spz) or Spz5 [17–19]. In order to bind to this receptor, pro-Spz must first be cleaved and activated downstream of specific extracellular serine protease (SP) cascades during embryogenesis or infection [17,20,21]. Upon infection, the terminal SP Spz-processing enzyme (SPE) cleaves Spz [22]. Two microbial recognition pathways initiate the SP cascades. One recognition pathway uses circulating PRRs

in the hemolymph to recognize pathogen-derived molecules, termed pathogen-associated molecular patterns (PAMPs). PRRs, such as Peptidoglycan recognition protein SA (PGRP-SA) and Gram-negative bacteria binding protein 3 (GNBP3), bind to specific PAMPs (Lys-type peptidoglycan and β-glucan, respectively), activating the initiator SP, modular serine protease (modSP), which further activates Gram-positive specific serine protease (Grass) [23–29]. Downstream of Grass, the SP paralogs Persephone (Psh) and Hayan are activated, leading to SPE activation [30]. The other recognition pathway relies upon Psh, and presumably Hayan, to recognize aberrant protease activity derived from pathogens in the hemolymph [27,29,30,31]. Specifically, Psh is cleaved by microbial proteases at its specific bait region, a region prone to cleavage by various proteases, leading to Psh activation and the induction of Spz cleavage by SPE [31,32]. Genetic analyses have revealed that sterile activation of the Toll pathway requires Spz and certain SPs, similar to its activation in response to infection. For example, clean injury to the larval epidermis activates the Toll pathway, depending on Spz and the SPs, Grass and SPE [6]. Likewise, clean injury to the embryo also requires Spz in order to activate the Toll pathway [4]. In apoptosis-deficient larvae lacking the initiator caspase Death regulator Nedd2-like caspase (Dronc), Spz and Psh are involved in Toll pathway activation [9]. In each context, the SPs involved in sterile activation of the Toll pathway have been identified. However, which SPs act as damage sensors in the cascades and which molecules from damaged sites serve as "danger signals" for activating these SPs remain unclarified.

Various molecules from damaged sites have been identified as danger signals that trigger sterile inflammation. In mammals, DAMPs from necrotic cells or extracellular matrix components bind to PRRs, such as TLRs, leading to the non-infectious activation of these receptors [2]. Another exemplary signal messenger frequently produced at injured sites is hydrogen peroxide ($H_2O_2$). Indeed, injury to tissues causes the activation of dual oxidase (Duox), which produces $H_2O_2$ [33,34]. Importantly, injection of $H_2O_2$ into the embryo activates the Toll pathway and its activation is dependent upon Spz [4]. Another study demonstrated that knockdown of Duox in the larval epidermis suppresses Toll pathway activation in the lymph gland upon epidermal injury [6]. These results suggest that the production of reactive oxygen species (ROS) at damaged sites likely triggers the initiation of the SP cascade, although the mechanisms remain unknown.

In *Drosophila* SP cascades, multiple paralogous SPs, such as Psh and Hayan, create redundancy, adding robustness to the activation of the Toll pathway. For example, a single mutation of either Psh or Hayan cannot completely suppress Toll pathway activation following septic injury [30]. Although SPE is considered the primary SP responsible for Spz cleavage during infection, a single mutation of SPE cannot fully suppress Toll activation under certain conditions of septic injury [30,35]. Yamamoto-Hino and Goto demonstrated that the overexpression of SP, Melanization Protease 1 (MP1), which has a phylogenetically close relationship with SPE, leads to the cleavage of Spz in *Drosophila* Schneider 2 (S2) cells [35]. However, similar to SPE, RNA interference of MP1 is insufficient to suppress Toll activation by septic injury [35]. Thus, the redundancy of the SPs responsible for Spz cleavage remains to be elucidated.

In this study, we examined the mechanism of sterile activation of the Toll pathway in an adult *Drosophila* model, where apoptosis deficiency in wing epidermal cells (WECs) induces Toll activation [10,11]. Genetic analyses using newly generated uncleavable *spz* mutant flies demonstrated that Spz cleavage is required for Toll activation in apoptosis-deficient flies. Proteomic analysis of adult hemolymph and a Spz cleavage assay within *Drosophila* S2 cells revealed that the SPs SPE and MP1 exhibit the highest Spz cleavage activity among the 13 hemolymph SPs and that Hayan and Psh are the upstream SPs of MP1 in S2 cells. Using genetic analysis, we found that loss of both Hayan and Psh, but not loss of both SPE and MP1, suppresses Toll pathway activation in apoptosis-deficient flies. When S2 cells overexpressing

pro-Psh are treated with $H_2O_2$, pro-Psh is activated and Spz is cleaved by MP1. As ROS production was detected in the necrotic wings, $H_2O_2$-mediated Psh activation may be involved in sensing damage in apoptosis-deficient flies. We collectively showed the crucial role of Hayan and Psh in the sterile activation of the Toll pathway in apoptosis-deficient flies. Furthermore, our findings highlight the importance of ROS as a signaling molecule for the damage-dependent activation of SPs such as Psh.

## Results

### Apoptosis deficiency of wing epidermal cells causes Toll pathway activation

During *Drosophila* development, WECs are cleared from the adult wings by undergoing apoptosis within 3 h after eclosion [36,37]. This apoptosis is mediated by a cysteine protease, caspase, that becomes activated upon making a protein complex, apoptosome [38]. In *Drosophila melanogaster*, Dronc (Mammalian orthologue is Caspase-9) and Death-associated APAF1-related killer (Dark) (Mammalian orthologue is Apaf-1) form the apoptosome [39]. Here, the caspase activation in WECs was blocked by knockdown of *Dark* using the QF/QUAS system, as reported in our previous studies [10,11]. Briefly, the *wing pouch (WP) -QF2* driver overexpresses a short hairpin RNA targeting *Dark* specifically in WECs by combination with *QUAS--Dark-sh*. This knockdown of *Dark* efficiently blocks apoptosis of WECs, leading to necrotic cell death characterized by cellular membrane permeabilization [10]. This transgenic fly ($WP^{QF}{>}Dark^{sh}$), denominated as an "apoptosis-deficient fly", has been known to exhibit innate immune responses such as the melanotic mass formation in wings and activation of the Toll and immune deficiency (IMD) pathways [10,11]. In apoptosis-deficient flies, activation of the IMD pathway is caused by dysbiosis of the gut microbiota, while activation of the Toll pathway is still observed in axenic conditions, implying that the Toll pathway responds to defects in apoptosis and not the presence of pathogens [11]. In apoptosis-deficient flies, expression levels of Toll target genes, including *Drosomycin* (*Drs*), *Bomanin Short 1* (*BomS1*), and *Bombardier* (*Bbd*), were approximately 50–60% of those in flies infected with *Micrococcus luteus*, a gram-positive bacterium (S1A–S1C Fig). First, we tested whether apoptosis defects in wings induced Toll pathway activation by determining whether removal of the wings would suppress Toll pathway activation in apoptosis-deficient flies. Wing excision suppressed the expression of *Drs*, a Toll-target AMP, in apoptosis-deficient flies to the level seen in the controls (Fig 1A). Furthermore, we examined the transition of Toll pathway activation after adult eclosion when reared on a standard diet or a diet supplemented with antibiotics (Fig 1B and 1C). In both conditions, although the expression level of *Drs* peaked 3–4 days after eclosion, it showed a decreasing trend 6 days after eclosion. Thus, these results demonstrate that apoptosis defects of the wings cause Toll pathway activation and that this activation is attenuated from 3–4 days after eclosion.

### Toll-mediated effector proteins are increased in the hemolymph of apoptosis-deficient flies

The Toll pathway elicits humoral immune responses, namely by secretion of effector proteins such as AMPs into the circulating hemolymph [16]. To further confirm the Toll pathway activation in apoptosis-deficient flies, we examined the differences in the hemolymph proteomes of the control ($WP^{QF}{>}+$) and apoptosis-deficient flies using proteomic analysis. In total, 861 proteins were identified in the hemolymph from both genotypes, of which 38.9% were predicted to have a signal sequence (annotated with "Signal" in UniProt Keywords) (Fig 1D and 1E). Among those with a signal sequence, 62 proteins were increased ($\log_2$ fold change $\geq 1.0$,

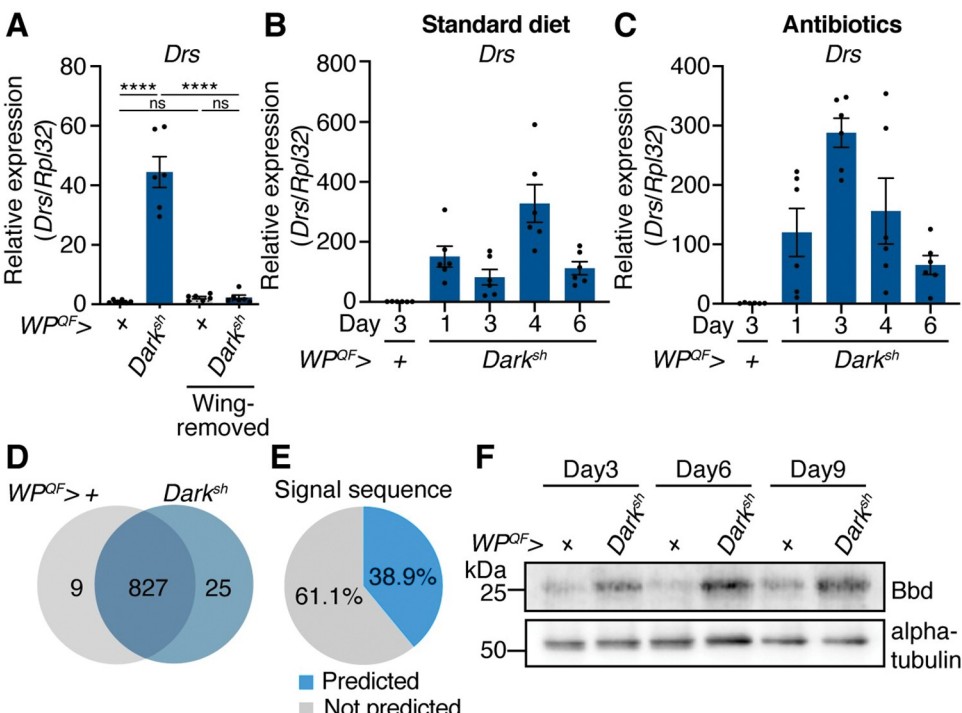

**Fig 1. Toll pathway is activated in apoptosis-deficient flies.** (A) Quantitative RT-PCR of *Drosomycin (Drs)* in the whole body of the control ($WP^{QF}$>+) and apoptosis-deficient ($WP^{QF}$>$Dark^{sh}$) male flies at 3 days after eclosion. Wings were removed 0–2 h after eclosion. n = 6. (B and C) Quantitative RT-PCR of *Drs* in the abdominal cuticles of control ($WP^{QF}$>+, 3 days after eclosion) and apoptosis-deficient ($WP^{QF}$>$Dark^{sh}$, 1–6 days after eclosion) male flies raised on a (B) standard or (C) antibiotics-supplemented diet. n = 6. (D and E) Proteomic analysis of adult hemolymph from control ($WP^{QF}$>+) and apoptosis-deficient ($WP^{QF}$>$Dark^{sh}$) male flies 6 days after eclosion. Three biological replicates were prepared for each genotype. (D) Venn diagram showing the number of identified proteins in the proteomic analysis of hemolymph from the control and apoptosis-deficient flies. (E) Pie chart showing the ratio of proteins having the putative signal sequence (annotated with "Signal" in UniProt Keywords) among the identified hemolymph proteins from the proteomic analysis. (F) Western blotting of the whole-body samples of the control ($WP^{QF}$>+) and apoptosis-deficient ($WP^{QF}$>$Dark^{sh}$) male flies against the Bombardier (Bbd) (anti-Bbd antibody) and alpha-tubulin (anti-alpha-tubulin antibody) proteins. Data are mean with SEM, and each dot represents a replicate in A–C. In A, statistical analysis was performed using one-way ANOVA with Tukey's multiple comparison test. ns: $P > 0.05$; ****: $P < 0.0001$.

adjusted *P* value < 0.1) in apoptosis-deficient flies (S2 Fig). Of these 62 proteins, 19 were encoded by core genes that are transcriptionally increased in response to most bacterial infections, hereafter referred to as "immune response core genes" [40,41]. We used i-*cis*Target [42,43] to identify the key regulators of the 62 upregulated proteins. Among the 62 proteins, 14 were encoded by genes whose regulatory regions contained a binding site for Dorsal-related immune factor (Dif), an NF-κB/Rel-like protein. Since Dif is the downstream transcription factor of Toll signaling, this analysis confirmed Toll pathway activation in apoptosis-deficient flies. Among the Dif target proteins, we further validated the increase in Bombardier (Bbd) protein, which is required for the delivery of Bomanins, a type of host defense peptides [44]. Expression of the *Bbd* gene is strongly induced through the Toll pathway upon infection or presence of other stimuli [40,45,46]. An increase in Bbd protein was confirmed by western blotting of samples from apoptosis-deficient flies at 3, 6, and 9 days after eclosion (Fig 1F). These results demonstrate that Toll pathway activation in apoptosis-deficient flies leads to the secretion of immune effector proteins, such as Bbd, into the hemolymph.

## Cleavage of Spz is required for Toll pathway activation in apoptosis-deficient flies

The Toll ligand Spz remains in a pro-form until it is activated through cleavage by SPs. Two SPs, Easter and SPE, cleave Spz during embryogenesis and infection, respectively [22,47]. To test the requirement of Spz and its cleavage in apoptosis-deficient flies, we generated uncleavable *spz* mutant flies (*spz^uc^*) using CRISPR/Cas9. The *spz^uc^* fly carries a mutated cleavage site composed of Leu-Asn instead of Arg-Val (Fig 2A), which protects Spz from being cleaved by SPE or Easter *in vitro* as indicated by a previous study [22]. During embryogenesis, the maternal *spz* of the *Drosophila* embryo is required for Toll activation to determine the dorso-ventral axis [48]; therefore, we first confirmed the phenotype of the *spz^uc^* mutant by observing its embryos. Consistent with previous studies [47,48], embryos from *spz^uc^* mothers did not hatch (Fig 2B) and showed a developmental defect characterized by a unique spätzle-like shape (Fig 2C). These phenotypes suggest that the mutation of the cleavage site functions *in vivo*. In the hemolymph of apoptosis-deficient flies, the C-terminal fragment of cleaved Spz (active Spz) was observed; however, the production of active Spz was blocked in the *spz^uc^* background (Fig 2D). Furthermore, we performed a transcriptome analysis of abdominal cuticles containing immune-responsive fat bodies, analogous to mammalian liver and adipose tissues. Compared to control flies ($WP^{QF}$>+), 212 genes were significantly upregulated in apoptosis-deficient flies ($WP^{QF}$>$Dark^{sh}$), of which 68 were suppressed in the *spz^uc^* background ($WP^{QF}$>$Dark^{sh}$, *spz^uc^*) (Fig 2E). Gene ontology (GO) analysis revealed that immune response genes were significantly enriched in these 68 genes (S1 Table). The i-*cis*Target analysis revealed the enrichment of Dif target genes in the 68 genes as well (S3 Fig). These data suggest that the upregulation of these 68 genes in apoptosis-deficient flies depends on Spz cleavage. Consistent with the transcriptome analysis, quantitative reverse transcription PCR (RT-PCR) showed that the upregulation of *Drs* in apoptosis-deficient flies was significantly suppressed in the *spz^uc^* mutant to the level observed in control flies (Fig 2F). We conclude that Toll pathway activation in apoptosis-deficient flies requires the cleavage of Spz to its active form via SPs.

## Exploration of hemolymph SPs capable of processing Spz

The requirement for Spz cleavage prompted us to explore the SPs responsible for processing Spz in the hemolymph of apoptosis-deficient flies. To identify SPs that can cleave Spz in an unbiased manner, we tested 13 hemolymph SPs identified by our proteomic analysis for their ability to cleave Spz in cells (Fig 3A and 3C). These 13 SPs included canonical extracellular components of the Toll pathway, such as SPE, Psh, modSP, and Hayan. Interestingly, Grass was not detected in our proteomic analysis, likely implying that the amount of Grass protein was relatively low compared to that of the other SPs in the hemolymph. We expressed the active form of each SP in *Drosophila* S2 cells by referring to the predicted structure of SPs [49]. The signal peptide from Easter was added to the N-terminus of each SP, as performed in a previous study [22], transporting them to the secretory pathway. Since S2 cells characteristically express Spz protein, the overexpressed active form of each SP should co-localize with the endogenous Spz and subsequently be able to cleave it in the secretory pathway [35]. The expression of each SP was confirmed by detecting the V5-tag signal in the cellular lysate (Fig 3B), and the signal of cleaved Spz was detected in the cellular lysates containing the active SPE, MP1, Psh, or Hayan (Fig 3C). Cleavage of Spz under the expression of active SPE confirmed that this assay could test the SP's ability to cleave Spz. As for MP1, it has previously been reported that overexpression of active MP1 in S2 cells leads to cleavage of Spz and that MP1 is closely related to Easter and SPE in its primary protein structure [35,49]. As the overexpression of active Psh or Hayan led to Spz cleavage, we hypothesized that active Psh and

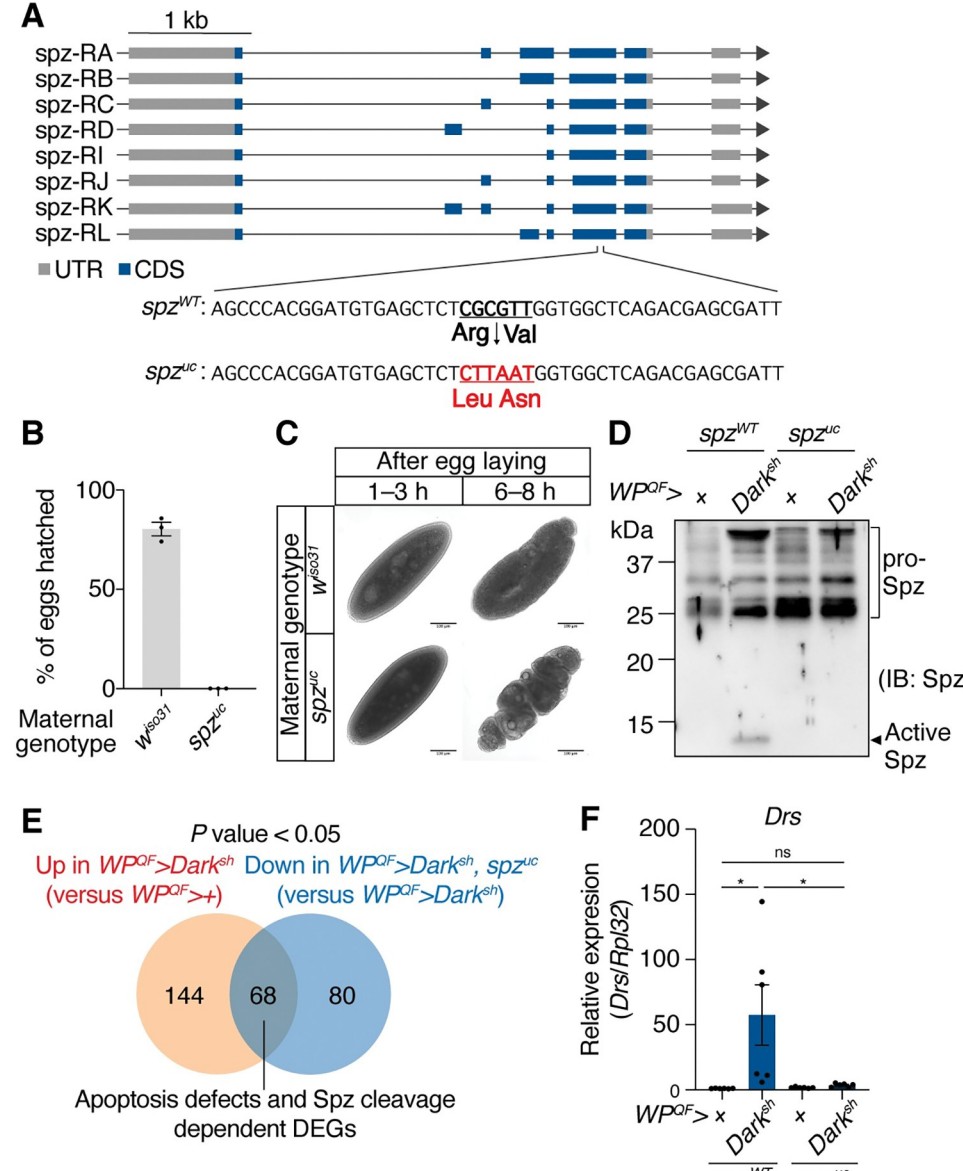

**Fig 2. Spätzle (Spz) cleavage is required for the Toll pathway activation in apoptosis-deficient flies.** (A) Schematic representation of the *spz* isoform variants and the mutation in the cleavage site. The cleavage site of wild-type *spz*, CGCGTT (Arg-Val) was mutated to the cleavage-resistant sequence CTTAAT (Leu-Asn) via CRISPR/Cas9. (B) Hatching rate of embryos from wild-type ($w^{iso31}$) or uncleavable Spz ($spz^{uc}$) female flies crossed with $w^{iso31}$ male flies. More than fifty embryos for each group. n = 3. (C) Representative images of embryos from wild-type ($w^{iso31}$) or $spz^{uc}$ female flies 1–3 h or 6–8 h after egg laying. Scale bars: 100 μm. (D) Western blotting of the hemolymph samples of the female flies against Spz (anti-Spz C106 antibody). Genotypes are described in the figure. (E) Venn-diagram showing the number of upregulated genes in apoptosis-deficient ($WP^{QF}>Dark^{sh}$) male flies (Fold change > 1.5, versus $WP^{QF}>+$) and the number of downregulated genes in apoptosis-deficient flies with the $spz^{uc}$ background ($WP^{QF}>Dark^{sh}, spz^{uc}$) (Fold change < 0.67, versus $WP^{QF}>Dark^{sh}$) with Benjamini–Hochberg adjusted *P* value < 0.05 after Wald test. (F) Quantitative RT-PCR of *Drs* in the whole body of the male flies 3 days after eclosion. n = 6. Genotypes are described in the figure. Data are mean with SEM, and each dot represents a replicate in B and F. In F, statistical analysis was performed using one-way ANOVA with Tukey's multiple comparison test. ns: $P > 0.05$; *: $P < 0.05$.

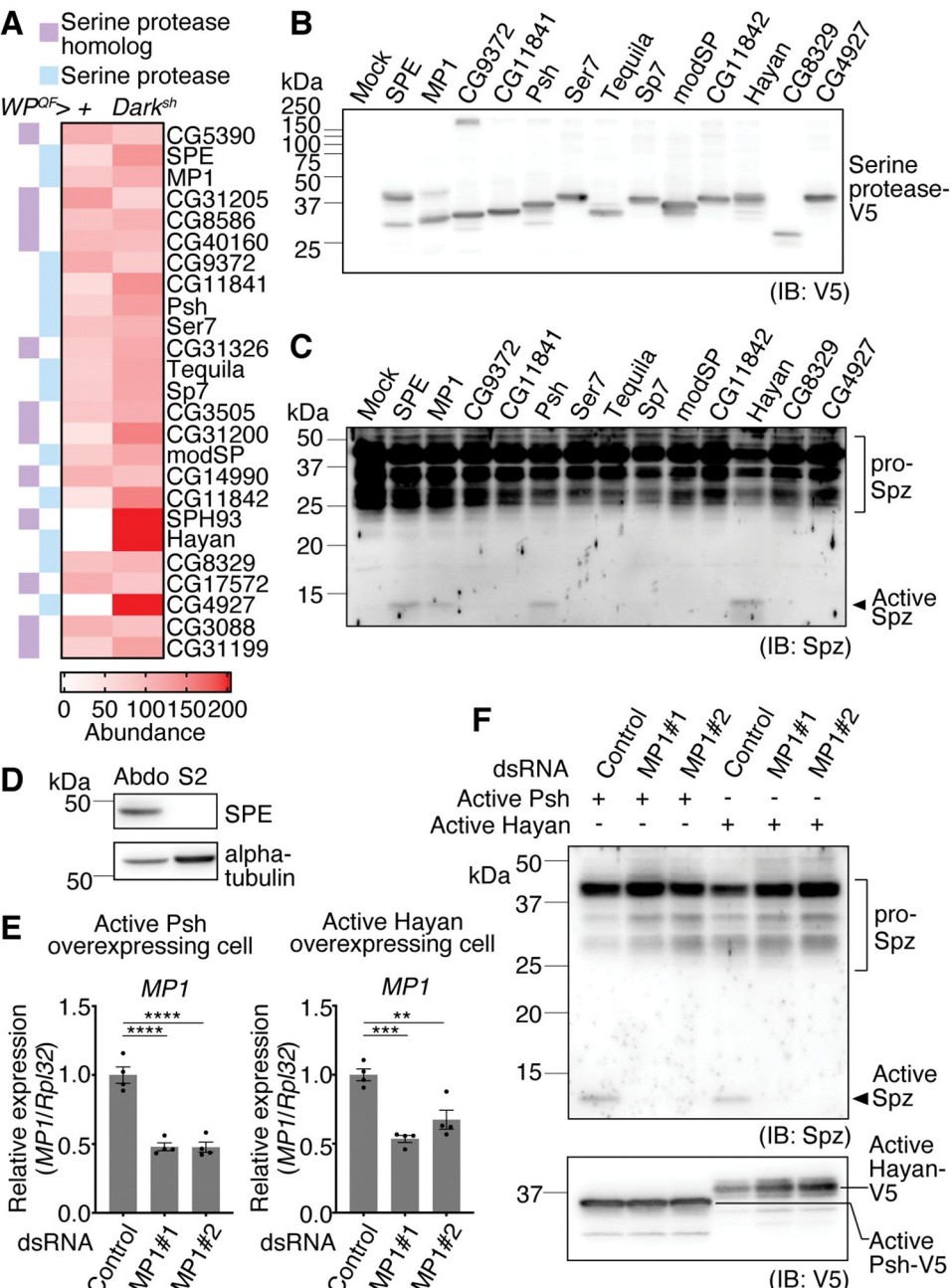

**Fig 3. Hayan and Persephone (Psh) are upstream of MP1 in Spz activation cascades in S2 cells.** (A) Heatmap showing the SPs and SP homologs having putative signal sequences identified in the hemolymph of the control ($WP^{QF}$>+) and apoptosis-deficient ($WP^{QF}$>$Dark^{sh}$) male flies by the proteomic analysis. This data is originated from the same proteome data of Fig 1D and 1E. (B and C) Western blotting of the lysates of S2 cells overexpressing the active form of each of the 13 SPs against (B) V5-tagged SPs (anti-V5 antibody) or (C) Spz (anti-Spz C106 antibody). (D) Western blotting of the abdominal cuticle sample (Abdo) and S2 cell lysates against SPE (anti-SPE antibody) and alpha-tubulin (anti-alpha-tubulin antibody). (E) Quantitative RT-PCR of *MP1* in S2 cells overexpressing the active form of Psh or Hayan with the treatment of each dsRNA (negative control, MP1#1, MP1#2). n = 4. Data are mean with SEM. Each dot represents a replicate. Statistical analysis was performed using one-way ANOVA with Tukey's multiple comparison test. **: $P < 0.01$; ***: $P < 0.001$; ****: $P < 0.0001$. (F) Western blotting of lysates of S2 cells overexpressing the active form of Psh or Hayan with the treatment of each dsRNA (negative control, MP1#1, MP1#2) against Spz (anti-Spz C106 antibody) and V5-tagged SPs (anti-V5 antibody).

Hayan activate endogenous SPs such as pro-SPE or pro-MP1 in S2 cells, leading to Spz cleavage. While the expression levels of *SPE*, *easter*, *Hayan*, and *psh* genes were markedly lower in S2 cells compared to those in whole-body fly samples, that of the *MP1* gene was comparable (S4A–S4E Fig). Consistent with the transcript level, SPE protein was not detected in the S2 cell lysate (Fig 3D); therefore, we knocked down MP1 in S2 cells (Fig 3E). Consistent with our hypothesis, the knockdown of MP1 suppressed Spz cleavage in S2 cells overexpressing active Psh or Hayan (Fig 3F). These results suggest that 1) among the identified hemolymph SPs, SPE and MP1 have a high capacity to cleave Spz, and 2) active Psh and Hayan require MP1 for Spz cleavage in S2 cells.

## Psh and Hayan are involved in the damage-sensing mechanisms of apoptosis-deficient flies

In S2 cells, Psh and Hayan act upstream of MP1, which upon activation cleaves Spz. The contribution of SPE, MP1, Psh, and Hayan to Toll pathway activation was then tested in apoptosis-deficient flies using SP mutants. For each genotype, flies with their wings removed were used as immunosuppressed controls. Specifically, we removed the wings from adult flies 0–18 h after eclosion and collected samples 3 days after eclosion. When the wings were removed, the expression of Toll target genes was suppressed in both $WP^{QF}>Dark^{sh}$ and $WP^{QF}>Dark^{sh}$ with $SPE^{SK6}$, $SPE^{SK6} + MP1^{SK6}$, $psh^{SK1}$, or $Hayan\text{-}psh^{Def}$ background, yet their expression remained higher in flies with $MP1^{SK6}$ or $Hayan^{SK3}$ background (Fig 4A–4F). Therefore, loss of MP1 or Hayan could have made the flies more responsive to the damage caused by wing removal via unknown mechanisms, promoting activation of the Toll pathway. Unexpectedly, $SPE^{SK6}$ or $MP1^{SK6}$ single mutants, as well as $SPE^{SK6} + MP1^{SK6}$ double mutants, showed Toll pathway activation in apoptosis-deficient flies that was comparable to (in $SPE^{SK6}$ and $SPE^{SK6} + MP1^{SK6}$) or even greater than (in $MP1^{SK6}$) that of the controls (Fig 4A–4C). In contrast, upon septic injury with *M. luteus*, the expression of Toll target genes was suppressed in $SPE^{SK6}$ but not in $MP1^{SK6}$ single mutants, and $SPE^{SK6} + MP1^{SK6}$ double mutants showed stronger suppression of Toll target genes than $SPE^{SK6}$ single mutants (S5A–S5C Fig). These results suggest that, although *SPE* and *MP1* redundantly contribute to Toll activation upon infection through Spz cleavage, they are not the SPs significantly involved in Spz cleavage in apoptosis-deficient flies. Additionally, $psh^{SK1}$ or $Hayan^{SK3}$ single mutants exhibited a comparable expression of Toll target genes to that of the controls, yet importantly, loss of both Hayan and Psh ($Hayan\text{-}psh^{Def}$) suppressed the expression of Toll target genes, with significant reduction in that of *BomS1* and *Bbd*, in apoptosis-deficient flies (Fig 4D–4F).

ModSP acts upstream of Hayan and Psh in the PRRs pathway that allows for microbial recognition [30]. Thus, the contribution of modSP was examined. Resultingly, $modSP^1$ mutant flies retained a similar level of Toll activation to that of controls (S6A–S6C Fig), indicating that the PRRs pathway does not significantly contribute to the activation of Hayan and Psh by apoptosis deficiency. In conclusion, paralogous SPs Hayan and Psh contribute to the sterile activation of the Toll pathway in apoptosis-deficient flies. Considering that the PRRs pathway does not play a major role in Toll activation in apoptosis-deficient flies, Hayan and Psh can be positioned at the starting point of the damage-sensing cascade and are activated by unknown cues from the damaged wings.

## Activation of Psh by $H_2O_2$ treatment in S2 cells leads to Spz cleavage

Previous studies have reported that $H_2O_2$ production at sites of injury contributes to sterile activation of the Toll pathway [4,6]. Next, we examined whether $H_2O_2$ contributes to the activation of Psh and Hayan. Putatively inactive pro-Psh or pro-Hayan (isoform A), along with

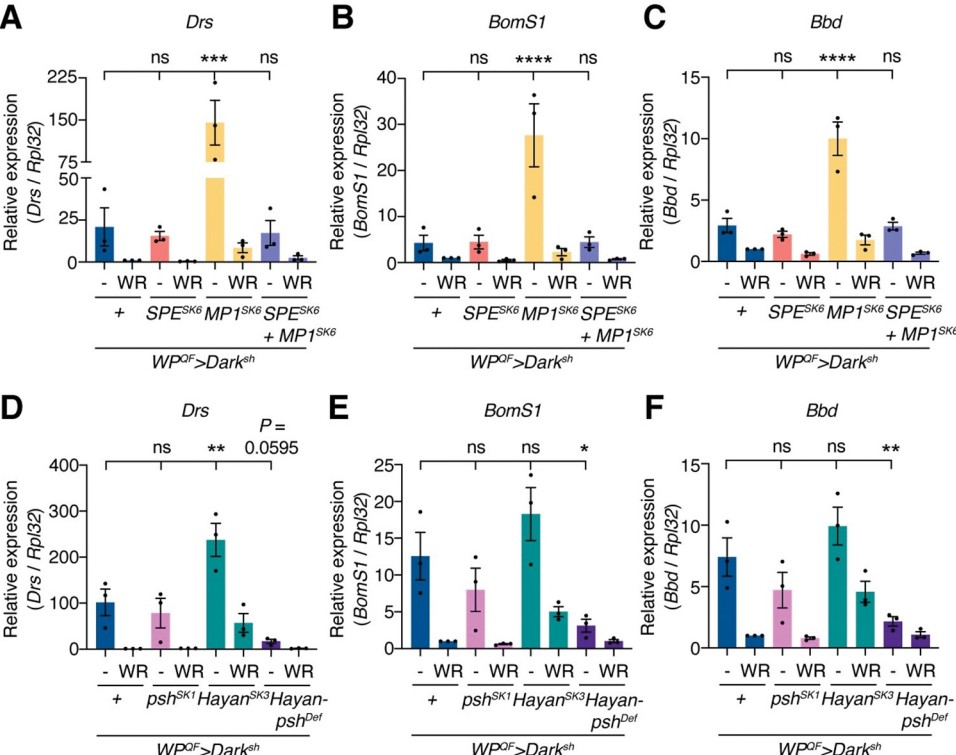

**Fig 4. Hayan and Psh are required for Toll pathway activation in apoptosis-deficient flies.** (A–F) Quantitative RT-PCR of *Drs* (A, D), *BomS1* (B, E), and *Bbd* (C, F) in the abdominal cuticles of apoptosis-deficient male flies or apoptosis-deficient male flies with SP-mutant backgrounds raised on an antibiotics-supplemented diet. WR: Wings Removed. Wings were removed 0–18 h after eclosion. n = 3. Data are mean with SEM. Each dot represents a replicate. Statistical analysis was performed using one-way ANOVA with Dunnett's multiple comparison test. ns: $P > 0.05$; *: $P < 0.05$; **: $P < 0.01$; ***: $P < 0.001$; ****: $P < 0.0001$.

Spz, were overexpressed in S2 cells, and Spz cleavage in cell lysates was examined when S2 cells were treated with $H_2O_2$. Overexpression of pro-Hayan led to Spz cleavage with or without $H_2O_2$ treatment (S7A Fig), suggesting that pro-Hayan is auto-activated by overexpression, similar to modSP [28]. A small amount of Spz was cleaved with the overexpression of pro-Psh, and this cleavage was strongly enhanced by approximately 4-fold upon treatment with $H_2O_2$ (S7B Fig). Notably, $H_2O_2$ treatment alone did not induce cleavage of Spz. Spz cleavage was not observed when cleavage-resistant Spz[uc] was overexpressed along with pro-Hayan or pro-Psh, indicating that Spz is processed at the canonical cleavage site to generate active Spz. When only pro-Psh was overexpressed, cleavage of endogenous Spz was also observed upon $H_2O_2$ treatment (S7C Fig). We also investigated whether overexpression of pro-Hayan or pro-Psh activates the Toll pathway *in vivo*. Each gene was overexpressed in adult fat bodies by using *Lpp-Gal4* in combination with *tub-Gal80[ts]*, where a temperature-sensitive Gal80 (Gal80[ts]) inhibits Gal4 at lower temperature (18°C). Overexpression of pro-Hayan, but not catalytic-dead pro-Hayan (pro-Hayan mutant; [50]) or pro-Psh, induced melanization in adult flies, as reported in a previous study using *Drosophila* larvae [50] (S8A Fig). Overexpression of pro-Hayan or pro-Psh, but not pro-Hayan mutant, activated the Toll pathway, while pro-Hayan induced stronger activation than pro-Psh (S8B–S8D Fig). This result is consistent with the results of experiments using S2 cells and support the idea that Hayan is more easily auto-activated than Psh when overexpressed.

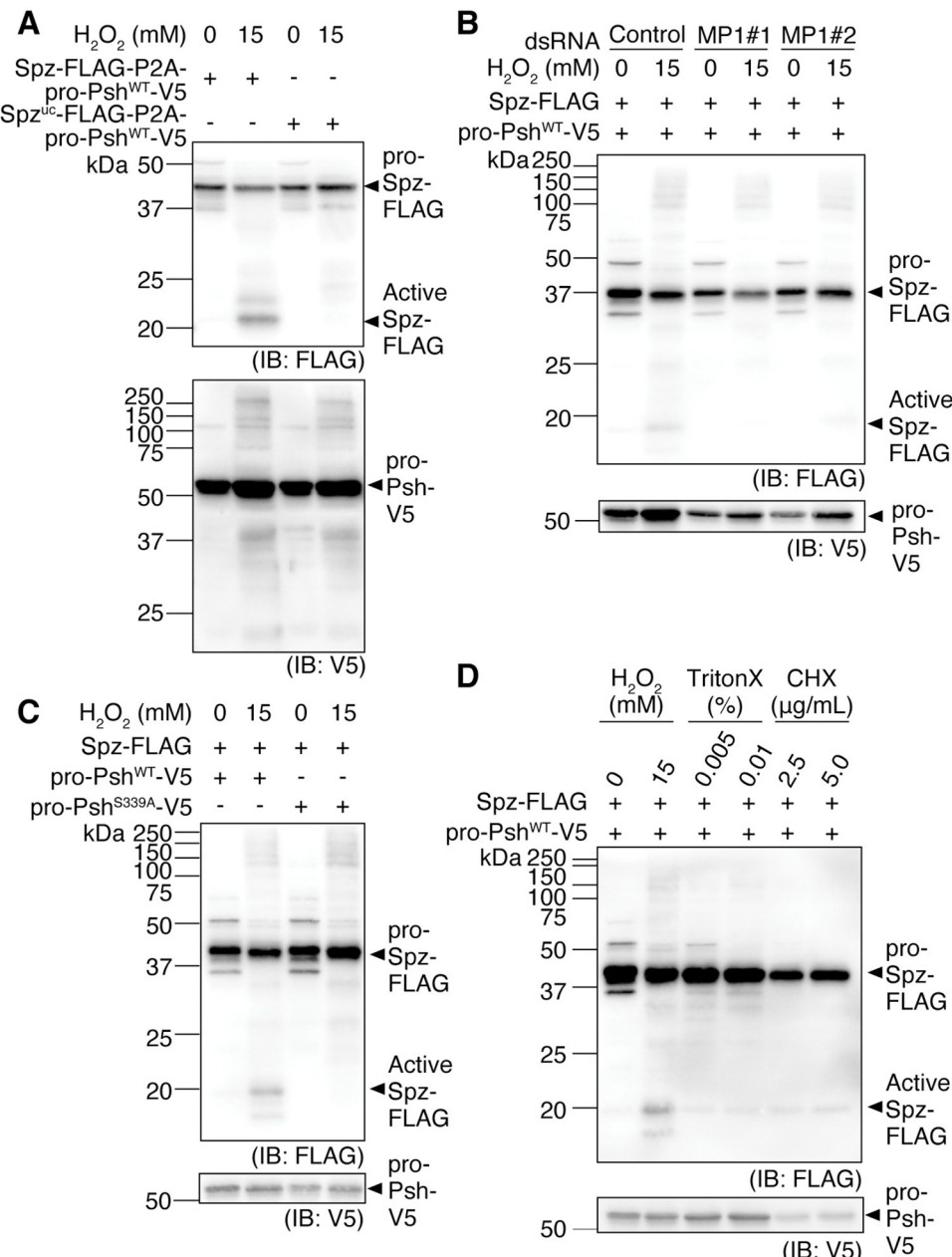

**Fig 5. Activation of pro-Psh by $H_2O_2$ treatment of S2 cells leads to Spz cleavage.** (A–D) Western blotting of S2 cell lysates against FLAG-tagged Spz (anti-FLAG antibody) and V5-tagged pro-Psh (anti-V5 antibody). (A) S2 cells were transfected with either *pMT-Spz-3xFLAG-P2A-pro-Psh^WT-V5* or *pMT-Spz^uc-3xFLAG-P2A-pro-Psh^WT-V5*. $H_2O_2$ treatment of 15 mM for 18 h. (B) S2 cells were transfected with *pMT-Spz-3xFLAG* and *pMT-pro-Psh^WT-V5*. dsRNA (negative control, MP1#1, MP1#2) treatment of 10 μg/mL for 2 weeks. $H_2O_2$ treatment of 15 mM for 18 h. (C) S2 cells were transfected with *pMT-Spz-3xFLAG* and *pMT-pro-Psh^WT or S339A-V5*. $H_2O_2$ treatment of 15 mM for 18 h. (D) S2 cells were transfected with *pMT-Spz-3xFLAG* and *pMT-pro-Psh^WT-V5*. $H_2O_2$ treatment of 15 mM, TritonX-100 treatment of 0.005% or 0.01% and cycloheximide (CHX) treatment of 2.5 μg/mL or 5.0 μg/mL for 18 h.

Moreover, we also tested Spz cleavage depending on pro-Psh and $H_2O_2$ treatment, by co-overexpressing Spz^WT or Spz^uc and pro-Psh using a single plasmid with the viral P2A sequence between the Spz^WT or Spz^uc and pro-Psh open reading frames (ORFs) [51,52] (Figs 5A and S7D). Cleavage of Spz^WT occurred depending on $H_2O_2$ treatment. Notably, in Fig 5A, Spz

cleavage was not observed when only overexpressing pro-Psh, unlike the result seen in S7B Fig. This difference is potentially caused by variation in the amount of overexpressed pro-Psh between samples. It is possible that when the amount of pro-Psh exceeds a certain threshold, Psh is also auto-activated to a small degree. After $H_2O_2$ treatment, several bands likely derived from pro-Psh were detected (Fig 5A, lower panel), indicating that the treatment induced pro-Psh processing and activation. Notably, a few other bands with larger molecular weights than that of pro-Psh existed. Activated Psh may be covalently attached to and inhibited by serpins, a family of SP inhibitors [53], leading to the fusion of Psh and a serpin to generate a protein of a larger molecular weight. Spz cleavage was suppressed when MP1 was knocked down by dsRNA treatment (Fig 5B). A Psh mutant containing a substitution of the Ser residue in the catalytic triad (Ser[339]) with Ala is catalytically inactive [32]. When Psh[S339A] was overexpressed, Spz cleavage was not observed, even after $H_2O_2$ treatment (Figs 5C and S7E), suggesting that Psh catalytic activity is required. These results demonstrate that the activation of Psh by $H_2O_2$ treatment further activates MP1, leading to Spz cleavage.

After incubation with $H_2O_2$ for 18 h, some S2 cells showed apoptotic traits, such as membrane blebbing, nuclear condensation, and cell membrane permeabilization (propidium iodide (PI)-positive), frequently observed in late-stage apoptosis or secondary necrosis (S9 Fig). To test whether apoptosis can activate Psh, we induced apoptosis in S2 cells by treatment with cycloheximide (CHX). At 18 h after CHX treatment, S2 cells showed apoptotic traits such as nuclear condensation and apoptotic body formation (S9 Fig). However, only a small amount of cleaved Spz was detected in the cell lysate (Fig 5D). Additionally, since $H_2O_2$ treatment also increased the number of cells with permeabilized plasma membranes, we tested whether cell membrane permeabilization was important for Psh activation. Although the number of PI-positive cells was increased by adding surfactant (TritonX-100) to the culture medium (S9 Fig), Spz cleavage was not observed, implying that cell membrane permeabilization alone was not sufficient to activate Psh (Fig 5D).

Upon infection, pro-Psh is activated in two steps [32]. First, pro-Psh is cleaved at the bait region by a wide range of microbial proteases, and then it is further processed between His[143] and Ile[144] by a circulating endogenous cysteine cathepsin, 26-29-p, to become activated. Gene expression levels of 26-29-p in S2 cells were comparable to those in whole-body fly samples (S10A Fig). To determine whether 26-29-p also contributes to Psh activation upon $H_2O_2$ treatment, 26-29-p was knocked down by dsRNA treatment of S2 cells. However, Spz cleavage was still observed (Fig 6A and 6B). In a previous study, it was reported that a mutated Psh bearing a substitution of His[143] with Glu[143] (Psh[H143E]) is resistant to cleavage by 26-29-p [32]. When pro-Psh[H143E] was overexpressed, $H_2O_2$ still induced Spz cleavage in S2 cells (S10B Fig). In addition, Spz cleavage was also observed when pro-Psh[H143G, or H143A] was overexpressed (Fig 6C). These results suggest that the canonical cleavage of Psh by 26-29-p did not significantly contribute to Psh activation upon $H_2O_2$ treatment of S2 cells.

Based on the structure predicted by AlphaFold [54,55], the canonical cleavage site of Psh (His[143] and Ile[144]) is located in a disordered region (18 amino acids from Ser[136] to Gly[153]) between a helix and the peptidase S1 domain (ref. AlphaFold DB: Q9VWU1). In general, disordered regions are susceptible to proteolysis [56,57]. To test the requirement of the disordered region for Psh-dependent Spz cleavage, pro-Psh carrying either a partial or complete deletion of the disordered region was overexperssed. For the partial deletions, 18 consecutive amino acids residues of the disordered region (Ser[136] to Gly[153]) were divided into three equal parts, and pro-Psh mutants carrying deletion of each 6 consecutive amino acids (Δ136–141, Δ142–147, or Δ148–153) were generated. When pro-Psh[Δ136–141] was overexpressed, Spz cleavage was still observed, while overexpression of pro-Psh mutants with the other partial deletions (pro-Psh[Δ142–147] or pro-Psh[Δ148–153]) or with the complete deletion (pro-Psh[Δ136–153]) did not

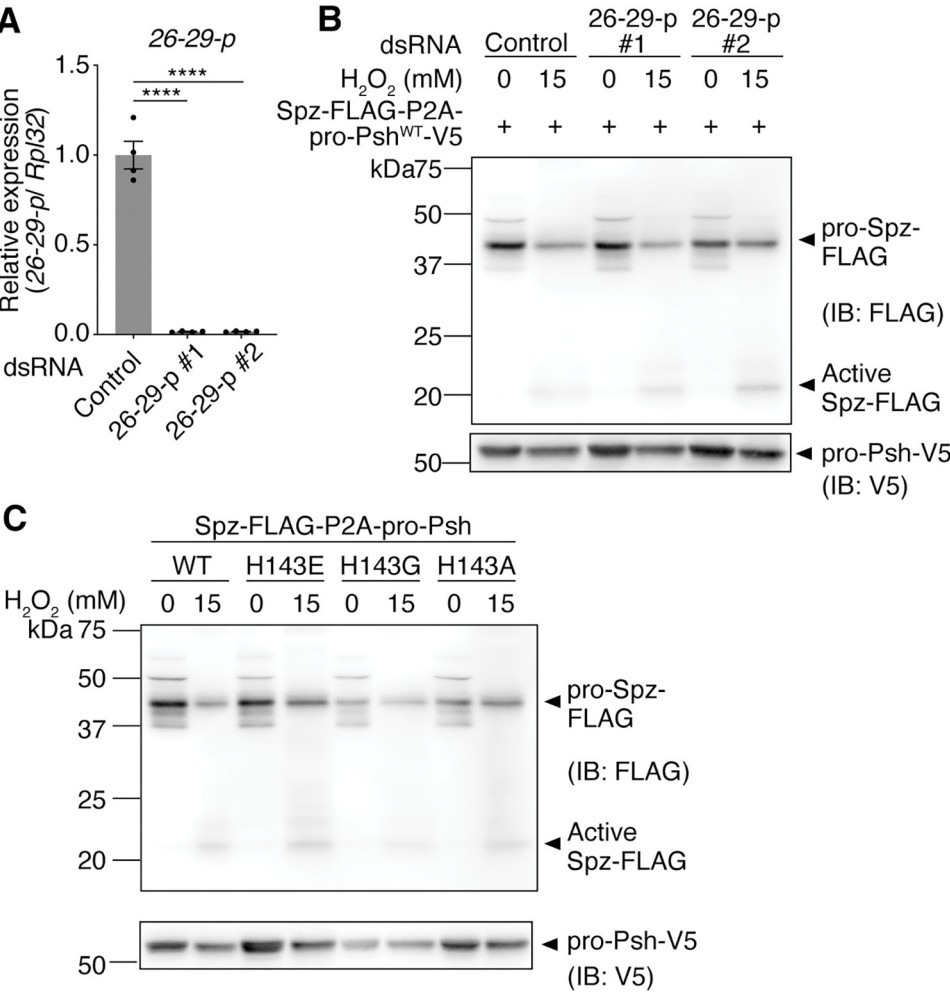

**Fig 6. Cathepsin 26-29-p does not significantly contribute to H$_2$O$_2$-Psh-dependent Spz cleavage.** (A) Quantitative RT-PCR of *26-29-p* in S2 cells treated with each dsRNA (negative control, 26-29-p#1, 26-29-p#2). n = 4. Data are mean with SEM. Each dot represents a replicate. Statistical analysis was performed using one-way ANOVA with Tukey's multiple comparison test. ****: *P* < 0.0001. (B and C) Western blotting of S2 cell lysates against FLAG-tagged Spz (anti-FLAG antibody) and V5-tagged pro-Psh (anti-V5 antibody). (B) S2 cells were transfected with *pMT-Spz-3xFLAG-P2A-pro-Psh*$^{WT}$*-V5*. dsRNA (negative control, 26-29-p#1, 26-29-p#2) treatment of 10 μg/mL for 2 weeks. H$_2$O$_2$ treatment of 15 mM for 18 h. (C) S2 cells were transfected with *pMT-Spz-3xFLAG-P2A-pro-Psh*$^{WT, H143E, H143G,}$ $^{or \; H143A}$*-V5*. H$_2$O$_2$ treatment of 15 mM for 18 h.

show Spz cleavage (S10C Fig). These results suggest that upon H$_2$O$_2$ treatment, the disordered region from Ile$^{142}$ to Gly$^{153}$ plays an important role in Psh activation, potentially by undergoing cleavage by intracellular proteases.

To elucidate ROS production in the wings of apoptosis-deficient flies, dihydroethidium (DHE) was injected into the flies and their wings were observed. In apoptosis-deficient flies 24–30 h after eclosion, the signal of oxidized DHE was detected in their wings and was co-localized with Hoechst signals from the DNA of remnant cells (Fig 7A). This result suggests that ROS production occurs in apoptosis-deficient wings in a similar manner to other damage conditions. In addition, when Catalase, a peroxide scavenger, was overexpressed in the apoptosis-deficient flies at a whole-body level, the expression levels of Toll target genes showed a decreasing trend (Fig 7B–7E). Taken together, these results demonstrate that Psh can be

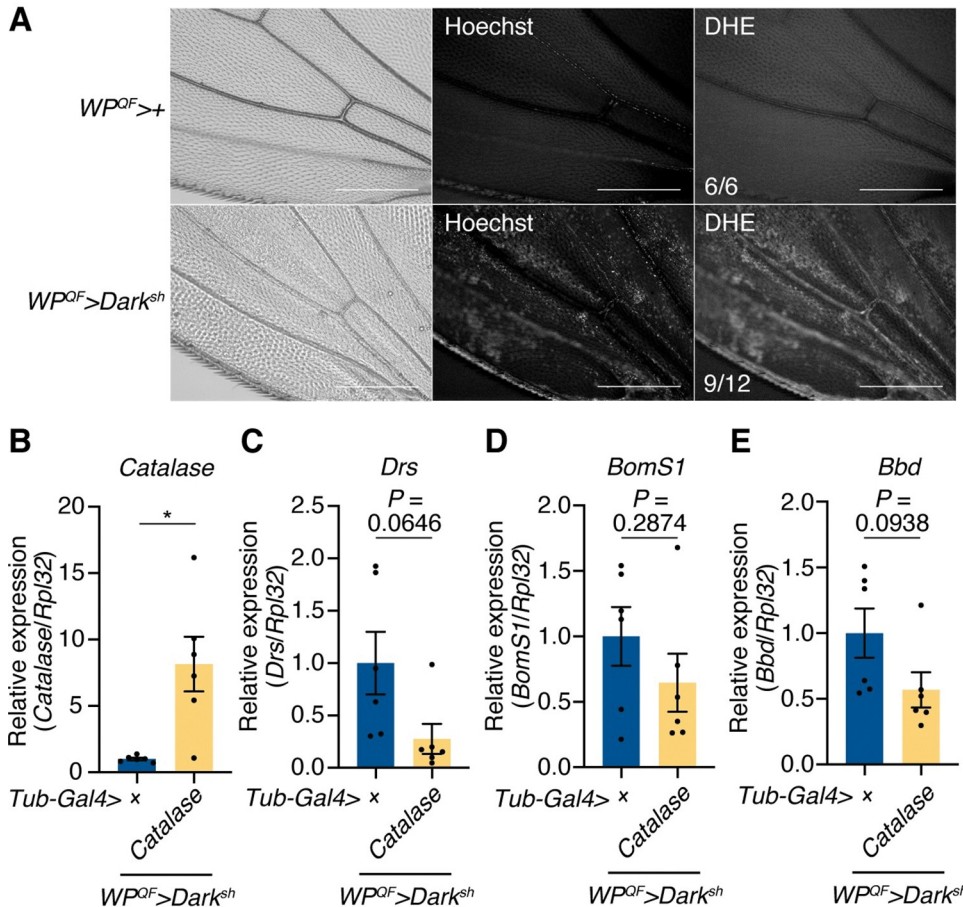

**Fig 7. ROS production is observed in wings of apoptosis-deficient flies.** (A) Representative images of wings of the control ($WP^{QF}>$) and apoptosis-deficient ($WP^{QF}>Dark^{sh}$) male flies 24–30 h after eclosion. Oxidized DHE signal indicates the production of reactive oxygen species (ROS). $WP^{QF}>+$, n = 6; $WP^{QF}>Dark^{sh}$, n = 12. Scale bars: 200 µm. (B–E) Quantitative RT-PCR of *Catalase* (B), *Drs* (C), *BomS1* (D), and *Bbd* (E) in the whole body of apoptosis-deficient male flies or apoptosis-deficient male flies with Catalase overexpression at the whole-body level raised on an antibiotics-supplemented diet. n = 6. Data are mean with SEM. Each dot represents a replicate. Statistical analysis was performed using two-tailed Welch's t test. *: $P < 0.05$.

activated in response to oxidative stimuli induced by $H_2O_2$ in S2 cells. Since ROS are detected in apoptosis-deficient wings and ROS scavenging reduces the expression of Toll target genes, it is likely that ROS production in the wings serves as a cue that triggers Psh activation in apoptosis-deficient flies.

## Discussion

In nature, tissue damage can increase the risk of secondary infection or lead to tissue dysfunction. Therefore, organisms must recognize and respond to damage to their body, often utilizing their innate immune systems to protect organismal homeostasis. In vertebrates, the molecular pattern recognition mechanism of immune-related receptors works as a powerful tool to recognize a wide range of molecules derived from site of injury. However, the molecular mechanism by which the innate immune system of *Drosophila* responds to damage is poorly understood. In this study, we addressed this question using the apoptosis-deficient *Drosophila* model, which exhibits common traits associated with tissue damage, such as cell necrosis and

ROS production in damaged areas. Furthermore, our molecular and genetic analyses provided insights into the various interactions and activation mechanisms of SPs.

In this study, we identified 13 hemolymph SPs in adult *Drosophila* using proteomic analysis and found that SPE and MP1 have higher capacities to cleave Spz than other hemolymph SPs. As the numerous SP genes in the *Drosophila* genome make it difficult to analyze SP cascades [30], focusing on the SPs and SP homologs existing in the hemolymph is a useful approach to understanding these signaling cascades. A previous study reported that overexpression of the active form of MP1 induces Spz cleavage [35]; however, the SP upstream of MP1 was not clearly determined. By combining molecular and biochemical approaches, we further clarified that MP1 acts downstream of Psh and Hayan in S2 cells. This discovery supports the effectiveness of our approaches at identifying unknown SP interactions, which can be difficult to reveal by *in vivo* genetic approaches alone. Previous studies assumed that some SPs are redundantly involved in processing Spz upon infection [30,35]. Given that MP1 exists in the adult hemolymph (according to our proteomics data) and that MP1 acts downstream of Hayan and Psh, MP1 would be involved in Spz cleavage *in vivo*, performing a function redundant to that of SPE. Indeed, we found that *SPE*, *MP1* double mutants show a stronger suppression of Toll target genes upon infection than *SPE* single mutants, providing further evidence of the redundant function of MP1 for Spz cleavage upon infection. In contrast, *SPE*, *MP*1 double mutants showed Toll activation similar to controls in apoptosis-deficient flies, implying that *SPE* and *MP*1 are not the significant SPs involved in Spz cleavage in this case. These results raise the question of how Spz cleavage occurs in apoptosis-deficient flies if SPE and MP1 do not play an important role. One hypothesis is that another hemolymph SP, other than SPE and MP1, cleaves Spz downstream of Hayan and Psh. Potentially, the proteomic analysis could not identify SPs present in small amounts.

Based on our results, Toll activation in apoptosis-deficient flies relies upon Hayan and Psh. If activation of Hayan and Psh occurs in the hemolymph of apoptosis-deficient flies, subsequent Spz cleavage by SPE and MP1 should occur in the hemolymph as seen in infection. However, the *SPE*, *MP1* double mutations did not impair Toll activation in apoptosis-deficient flies. Thus, another proposed mechanism could be that Spz cleavage that depends on Hayan and Psh occurs intracellularly in response to necrotic damage. This hypothesis could be indirectly supported by the following evidence. First, in the present study, pro-Hayan overexpression or pro-Psh overexpression with $H_2O_2$ treatment induces intracellular Spz cleavage in S2 cells. Moreover, upregulation of *Hayan* gene expression occurs in hemocytes upon epidermal injury [5]. In addition, a previous study reported that $H_2O_2$ produced at wound sites is efficiently diffused into hemocytes through Prip, a channel protein [5]. Thus, these support the idea that, in Spz-producing cells, pro-Hayan or pro-Psh can be intracellularly activated in response to tissue damage. Although MP1 is required for Spz cleavage in S2 cells, it is possible that *in vivo* another SP becomes activated downstream of Hayan or Psh to cleave Spz in hemocytes recruited to damaged sites or in damaged cells themselves. We found that in wings of apoptosis-deficient flies, expression levels of *spz*, *Hayan*, and *psh* genes were higher than those in whole-body flies or S2 cells (S11A–S11C Fig). In contrast, gene expression levels of *Hemese*, a hemocyte marker, were lower in the wings than those in whole-body flies or S2 cells, indicating that the number of recruited hemocytes is not significant in the wings (S11D Fig). Based on these results, it is possible that Spz cleavage downstream of Hayan and Psh occurs in apoptosis-deficient WECs rather than in recruited hemocytes. We also found that ηTrypsin, a *Drosophila* trypsin, could induce Spz cleavage in S2 cell lysates (S12A–S12C Fig), suggesting that certain trypsin-like SPs can cleave Spz intracellularly.

Furthermore, it has been reported that Spz belongs to the neurotrophin protein family, sharing the conserved cystine-knot structure with mammalian neurotrophins, such as brain-

derived neurotrophic factor (BDNF) or nerve growth factor (NGF) [47,58,59]. In mammals, BDNF is synthesized as the precursor pro-BDNF and matures by protease-mediated cleavage, a process reminiscent of pro-Spz activation [60]. Importantly, it has been reported that processing of pro-BDNF not only occurs extracellularly via proteases such as plasmin, but also occurs intracellularly in secretory pathways (trans-Golgi network and secretory vesicles) via furin and proprotein convertases [61–63]. Given the similarity between Spz and BDNF, Spz may also be cleaved intracellularly to achieve maturation in response to damage. While this hypothesis is still at a conceptual stage, future investigation into whether Spz cleavage also occurs intracellularly upon tissue damage *in vivo*, along with which proteases are involved, is worthwhile.

Our finding that Hayan and Psh contribute to Toll pathway activation in apoptosis-deficient flies highlights their importance in the damage-sensing mechanisms of adult *Drosophila*. In apoptosis-deficient larvae, Toll pathway activation can be suppressed by a single *psh* mutation [9], potentially reflecting differences in isoforms or amount of Hayan expressed in larvae compared to those in adults (FlyAtlas 2: www.flyatlas2.org, [64,65]). However, there seems to be a discrepancy in the requirement of Psh for sterile activation of the Toll pathway because Toll activation upon clean injury to the larval epidermis is not suppressed in *psh* single mutants [6]. Hayan may redundantly contribute to this Toll activation upon larval injury because Hayan is involved in melanization upon injury [30,50]. Since the redundant functions of Hayan and Psh in Toll activation has been already established in a previous study [30] and confirmed in the present study, investigation of whether Toll pathway activation upon larval injury is also suppressed in *Hayan* and *psh* double mutants is necessary.

In this study, we could not determine how Hayan and Psh are activated in response to the damage caused in apoptosis-deficient flies; however, the experiments with S2 cells may provide clues to these unknown mechanisms. Overexpression of pro-Hayan (isoform A) in S2 cells resulted in Spz cleavage, suggesting that pro-Hayan can be auto-activated, an activation mode reminiscent of that of the initiator protease modSP [28]. In our proteomic analysis, Hayan protein was undetectably low in control flies, yet was increased in apoptosis-deficient flies. This increasing trend was also observed in a previous study that reported upregulation of the *Hayan* gene in hemocytes upon clean injury to adult *Drosophila* [5]. These observations likely support the idea that more Hayan is produced upon tissue damage, and this increased Hayan is auto-activated like an initiator protease to trigger Toll activation and melanization in damaged sites.

One of the most important findings regarding Psh is that the pro-Psh overexpressed in S2 cells was activated upon treatment with $H_2O_2$. Previous studies have demonstrated that clean injury leads to Toll activation in response to $H_2O_2$ production in the injured epidermis, proposing a potential mechanism through which $H_2O_2$ activates SPs [4,6]. Thus, the $H_2O_2$-dependent activation of Psh observed in this study might be a common damage-sensing mechanism of *Drosophila*. However, the mechanism by which $H_2O_2$ induces Psh activation in cells remains to be determined. A candidate cysteine cathepsin 26-29-p, which can process and activate Psh upon infection [32], was not significantly involved in Psh activation induced by $H_2O_2$ treatment of S2 cells. We also found that Psh mutants carrying deletion of the disordered domain between the CLIP- and peptidase S1 domains did not show Spz cleavage upon $H_2O_2$ treatment. Since the unfolded disordered domain is susceptible to proteolysis [56,57], Psh is likely activated through cleavage within this domain by unknown proteases other than 26-29-p downstream of $H_2O_2$. Since Psh is translocated into the endoplasmic reticulum, it is possible that Psh activation in S2 cells occurs in the secretory pathway. Therefore, it is necessary to investigate the involvement of proteases in Psh processing through the secretory pathway.

It is also possible that $H_2O_2$ directly activates Psh by oxidation of amino acid residues such as methionine or cysteine. For example, $H_2O_2$ can oxidize AMP-activated protein kinase

(AMPK) alpha subunit at its cysteine residue, activating AMPK [66]. In another way, the oxidative property of $H_2O_2$ may contribute to promoting Psh proteolytic activity by inhibiting negative regulators. For instance, the oxidation of protease inhibitors such as alpha-1-antitrypsin, a member of the serpin family, and alpha-2-macroglobulin inhibits their proteinase inhibitory activity, consequently, promoting proteolysis by active proteinases (reviewed in [67]). In *Drosophila*, serpins such as *necrotic* play a role in suppression of the Toll pathway activation by inhibiting the SP cascade [68]. Therefore, inhibition of serpins by $H_2O_2$ can likely accelerate proteolysis by active Psh in cells and *in vivo*. These hypothesized $H_2O_2$-mediated Psh activation mechanisms are worth investigating in future studies, potentially extending our understanding of the damage-sensing mechanism of *Drosophila*.

The present study indicates that the Psh-Spz cleavage cascade initiated upon necrotic damage is mediated by proteases other than those SPs known to be involved in canonical SP cascades. Thus, the unbiased study of potential mediator proteases is a critical area for future research. Initially, we performed an RNAi screening of SPs. The UAS-RNAi fly strain for each SP and SP-related gene was individually crossed with flies carrying the whole-body driver *Da-Gal4*, which had Q system-based apoptosis defects in the wings. A total of 22 RNAi lines showed mortality or wing defect phenotypes caused by *Da-Gal4*, and 85 RNAi flies were subjected to quantitative RT-PCR analysis for *Drs* expression (S2 Table). Although RNAi screening alone does not conclusively identify candidate genes owing to knockdown efficiency and possible off-target effects, the resulting list of candidate genes may still aid in further identification of the responsible SPs. For example, the contribution of Hayan and Psh was implied by the result of our RNAi-based screening. In this analysis, many trypsin family genes were also identified. Similar to how the treatment of embryos with trypsin activates the Toll pathway [4], we observed that treatment of pro-Spz-expressing S2 cell lysate with a *Drosophila* trypsin, ηTrypsin, can generate active Spz (S12A–S12C Fig). Thus, trypsin family genes may be directly or indirectly involved in pro-Spz activation under noninfectious conditions.

In conclusion, apoptosis defects in WECs lead to sterile activation of the Toll pathway through Hayan and Psh, establishing them as sensors for necrotic damage (S13 Fig). Given that *SPE*, *MP1* double mutant flies did not show impairment of Toll activation in apoptosis-deficient flies, it is likely that proteases other than SPE and MP1 also cleave Spz downstream of Hayan and Psh. Our data suggest that Hayan and Psh can be activated putatively in the secretory pathway of S2 cells. These results raise the conceptual idea that upon necrotic damage, activation of Hayan and Psh occurs in the secretory pathway of cells, promoting intracellular Spz cleavage mainly depending on proteases other than SPE and MP1. The ROS production observed in necrotic wings may contribute to the intracellular activation of Psh. In the case of tissue damage, the Toll pathway may not be activated by DAMPs, but instead by damage-associated signaling molecules (DASMs) such as ROS. Since ROS production is a frequent characteristic observed at site of injury, it can be proposed that ROS-mediated SP activation is the common mechanism that initiates the *Drosophila* Toll pathway activation in response to general tissue damage.

## Materials and methods

### Fly stocks and rearing conditions

Files were raised on a standard *Drosophila* diet (4% cornmeal, 6% baker's yeast (Saf yeast), 6% glucose, and 0.8% agar with 0.3% propionic acid and 0.15% nipagin) at 25°C unless otherwise indicated. For the experiments shown in Figs 1C, 4A–F, 7B–7E, and S6A–S6C, flies were raised at 25°C on an antibiotic-supplemented diet (the standard diet with 0.21% nipagin, 200 μg/mL rifampicin, 50 μg/mL tetracycline, 500 μg/mL ampicillin), by referring to a previous study

[11]. The $w^{iso31}$ strain developed in a previous study was used as the wild-type strain [69]. Additionally, the following fly strains were used in this study: *WP-QF2*, *QUAS-Dark$^{sh}$* on the 2$^{nd}$ chr., and *QUAS-Dark$^{sh}$* on the 3$^{rd}$ chr. (generated in our laboratory in previous studies [11,70]), *SPE$^{SK6}$* (kindly provided by Dr. M. Yamamoto-Hino and Dr. S. Goto), *MP1$^{SK6}$*, *psh$^{SK1}$*, *Hayan$^{SK3}$*, *Hayan-psh$^{Def}$*, and *modSP$^1$* (kindly provided by Dr. B. Lemaitre), *Tub-Gal4* (DGRC 108069, obtained from Kyoto *Drosophila* Stock Center, Japan), *UAS-Catalase* (BL 24621, obtained from Bloomington *Drosophila* Stock Center, U.S.A.), *Lpp-Gal4* (kindly provided by Dr. M. Brankatschk and Dr. S. Eaton, [71]), *UAS-LacZ* (DGRC 106500, obtained from Kyoto *Drosophila* Stock Center, Japan), *UAS-persephone* (kindly provided by Dr. JM. Reichhart, [31]), *UAS-Hayan* and *UAS-Hayan mutant* (kindly provided by Dr. WJ. Lee, [50]).

## Infection experiments with *Micrococcus luteus*

*Micrococcus luteus*, kindly provided by Dr. M. Yamamoto-Hino and Dr. S. Goto, was cultured in 30 mL lithium borate (LB) buffer (12780052, Thermo Fisher Scientific, U.S.A.) at 29˚C, and the culture's optical density (OD) at 600 nm was measured. The culture medium of OD 0.4–0.5 was centrifuged at $20,000 \times g$ for 3 min at 4˚C, and the supernatant was discarded. The pellet was resuspended in 50 μL of phosphate-buffered saline (PBS) to form the *M. luteus* medium. Adult male flies 3 days after eclosion were pricked in the thorax using a metal needle dipped in 70% ethanol and then in the *M. luteus* medium to induce septic injury. For clean injury, adult flies 3 days after eclosion were pricked in the thorax by a metal needle dipped in 70% ethanol only. After 24 h, treated (septic injury and clean injury) and untreated control flies were collected for RNA extraction.

## Generation of mutant with uncleavable *spz*

The uncleavable *spz* mutant (*spz$^{uc}$*) carries a mutation in the Spz cleavage site (3R:27066239–27066244), where "CGC-GTT (Arg-Val)" is changed to "CTT-AAT (Leu-Asn)" [22]. This mutation was knocked in using CRISPR/Cas9-mediated homology directed repair (HDR) [72]. A pair of guide RNAs (gRNAs) was designed using the CRISPR optimal target finder tool available on flyCRISPR (http://flycrispr.molbio.wisc.edu/). The primers for the gRNA were annealed and subcloned into the BbsI-digested U6b-sgRNA-short vector ([73]; kindly provided by Dr. N. Perrimon) using the Mighty Mix DNA Ligation Kit (6023, Takara Bio, Japan). The following primers were used to construct a pair of gRNA vectors: spz$^{uc}$ gRNA vector-1, 5'-TTCGTCCACCAGGTACTCGCTGAT-3' and 5'-AAACATCAGCGAGTACCTGGTGGA-3'; spz$^{uc}$ gRNA vector-2, 5'- TTCGGCTTATGTGTGAGGCCTGAA-3' and 5'-AAACTTCAGGCCTCACACATAAGC-3'.

To generate the HDR donor template, three DNA fragments of the 5' homology arm with the mutated cleavage site were sub-cloned into the BsaI site, and one DNA fragment of the 3' homology arm was sub-cloned into the SapI site of pBac[3xP3-DsRed_polyA_Scarless_TK] (generated by Dr. T. Katsuyama, refer to [72]) using the In-Fusion HD Cloning Kit (639650, Takara Bio, Japan). Protospacer adjacent motif (PAM) sequences of the gRNA-binding sites in the donor template were mutated to block Cas9-mediated cleavage after HDR. The primers were as follows: 5' homology arm-1 with mutated PAM, 5'-TGAAGGTCTCCTTAACGAC-GATCCGACCCAGAAAC-3' and 5'-CCTGAAAGATCCGGATAGTCGTCCACATTC-3; 5' homology arm-2 with mutated PAM and the mutated cleavage site, 5'- CTATCCGGATCTTT CAGGCCTCACACATAAG-3' and 5'- CCATTAAGAGAGCTCACATCCGTGGGCT-3; 5' homology arm-3 with the mutated cleavage site, 5'-GAGCTCTCTTAATGGTGGCTCAGAC GAGCGATT-3' and 5'- TCTTTCTAGGGTTAATTCAAAGAACTTTACTTTGC-3; and 3' homology arm with mutated PAM, 5'-

TCTTTCTAGGGTTAAAACATCAGCGAGTACCTGGT-3' and 5'-GACGGCTCTTCATTA GGAGATATATGGATAGTTGC-3'. The constructed vectors were injected into the embryos of *y[1] M{GFP[E.3xP3] = vas-Cas9. RFP-}ZH-2A w[1118]* (BDSC#55821) in BestGene Inc. (State of California, U.S.A.). Each Ds-Red-positive transformant was isogenized and confirmed by genomic PCR and sequencing. The following primers were used for genomic PCR: fragment 1, 5'-CATAGCCAAGTATCGGCCAC-3' and 5'-GTCTCTAATTGAATTAGAT CCCG-3'; fragment 2, 5'-CACTGCATTCTAGTTGTG-3' and 5'-CAGGAAGCCGAAGTCG ATGCATG-3'. To avoid background effects, the generated strains were backcrossed for six generations to $w^{iso31}$.

## Embryo observation

$w^{iso31}$ or $spz^{uc/uc}$ female flies were mated with $w^{iso31}$ male flies. Embryos were collected for 2 h using acetic acid agar plates (2.3% agar, 1% sucrose, and 0.35% acetic acid) and incubated in a 25°C incubator for an additional 1 or 6 h. Embryos were collected from the plates into baskets covered with mesh fabric. Collected embryos were fixed as follows. Briefly, the embryos were incubated for 3 min in 3% sodium hypochlorite to achieve dechorionation. Embryos were washed with Milli-Q water and collected in a 2-mL tube containing 1 mL heptane saturated with 4% paraformaldehyde (PFA) in PBS. Next, 1mL of 4% PFA in PBS was added, and the tubes were vigorously shaken for 15 s, followed by incubation on a shaker at room temperature for 20 min. The lower PFA layer was removed, and 1 mL of methanol was added. After vigorously shaking the tubes for another 15 s, the upper heptane layer was removed. The remaining solution was reduced to 500 μL, and 500 μL of PBS with 0.3% TritonX-100 (PBST) was added. This process was repeated three more times to replace the methanol with 0.3% PBST. Embryos were mounted with 80% glycerol/Milli-Q water. Images were obtained using a Leica TCS SP8 confocal microscope (Leica Microsystems, Germany).

## Measurement of hatchability

Once again, $w^{iso31}$ or $spz^{uc/uc}$ female flies were mated with $w^{iso31}$ male flies, and embryos were collected for 6 h using acetic acid agar plates. After the plates were collected, more than 50 eggs were transferred to another agar plate and incubated for 24 h in a 25°C incubator. After 24 h, the number of hatched eggs was measured, and the hatchability rate was calculated as the percentage of the eggs that hatched.

## Sample preparation for LC-MS/MS analysis

To collect adult hemolymph from the flies, we followed the method described in a previous study [74] with some modifications. Male flies (80–100) were anesthetized with $CO_2$. To wash out contaminated food and feces, flies were transferred to a 1.5 mL centrifuge tube and washed as follows: flies were washed with 1 mL of PBS with 0.1% Triton X-100 (PBST) →twice with 1 mL of Milli-Q water → twice with 1 mL of 50% ethanol → with 1 mL of 50 mM ammonium bicarbonate (AMBC) in ultrapure water. The washed flies were transferred to a petri dish and placed on ice. Next, 150 μL of 50 mM AMBC was dropped on the parafilm, and the middle leg on the right side was removed from each fly in the drop. The whole drop containing all treated flies was transferred to a filter device (Ultrafree-MC, GV 0.22 μm (UFC30GVNB, Merck, Germany)), and an additional 150 μL of 50 mM AMBC was added (in total 300 μL of AMBC and treated flies). The device was centrifuged at 2,000 rpm for 2 min at 4°C, and the flow-through was transferred to the same filter device. The device was centrifuged again at 2,000 rpm for 2 min at 4°C, and the flow-through was transferred to a 1.5 mL Protein LoBind Tube (0030108116, Eppendorf, Germany), flash-frozen, and stored at -80°C until further use.

Hemolymph samples were thawed on ice, and protein concentrations were measured using a BCA Protein Assay Kit (297–73101, FUJIFILM Wako, Japan). Tris(2-carboxyethyl)phosphine (TCEP; C4706, Sigma-Aldrich, U.S.A.) (50 mM) was added to each sample containing 20 μg protein to a final concentration of 5 mM before incubation at 60°C for 30 min. Methyl methanethiosulfonate (MMTS; 23011, Thermo Fisher Scientific, U.S.A.) (200 mM) was added to a final concentration of 10 mM before incubation at room temperature for 30 min in the dark. Sequencing Grade Modified trypsin (V5111, Promega, U.S.A.) was added at a 1:20 ratio, and the samples were digested at 37°C overnight. Next, 10% trifluoroacetic acid (TFA; 206–10731, FUJIFILM Wako, Japan) was added to a final concentration of 0.5% for acidification before incubation at 37°C for 1 h. The samples were centrifuged at 20,000 × $g$ for 10 min at 4°C, and the supernatants were purified using GL-Tip SDB/GC (7820–11200, GL Sciences, Japan) according to the recommended protocol of the manufacturer with modifications. Briefly, the rinsing process was performed twice with 100 μL of solution A (0.1% TFA, 5% acetonitrile; 015–08633, FUJIFILM Wako, Japan), and the peptides were eluted with 50 μL of solution B (0.1% TFA, 80% acetonitrile). The purified samples were dried using a speed-vac and resuspended in 15 μL of 0.1% formic acid-distilled water (16245–63, Kanto Chemical, Japan). Protein concentrations were measured using the BCA Protein Assay Kit, and the samples were diluted with 0.1% formic acid-distilled water to 200 ng/μL. The samples were centrifuged at 20,000 × $g$ for 10 min at 4°C, and the supernatants were collected. Purified samples were subjected to liquid chromatography with tandem mass spectrometry (LC-MS/MS) analysis.

## LC-MS/MS analysis

Samples were loaded onto an Acclaim PepMap 100 C18 column (75 μm × 2 cm, 3-μm particle size, and 100 Å pore size; 164946, Thermo Fisher Scientific, U.S.A.) and separated using a nano-capillary C18 column (75 μm × 12.5 cm, 3 μm particle size, NTCC-360/75-3-125, Nikkyo Technos, Japan) and an EASY-nLC 1200 system (Thermo Fisher Scientific, U.S.A.). The elution conditions are described in the S3 Table. The separated peptides were analyzed using QExactive (Thermo Fisher Scientific, U.S.A.) in data-dependent MS/MS mode. The MS/MS analysis parameters are described in the S4 Table. Collected data were analyzed using Proteome Discoverer (PD) 2.2 software with the Sequest HT search engine. The PD 2.2 analysis parameters are described in the S5 Table. Peptides were filtered at a false discovery rate of 0.01 using the Percolator node. Label-free quantification was performed based on the intensities of the precursor ions using the precursor ion quantifier node. Normalization was performed using the total peptide amount in all average scaling modes. The MS proteomics data were deposited in the ProteomeXchange Consortium via the jPOST partner repository with the dataset identifier PXD037066 [75]. The analysis of putative transcription factor-binding sites in the gene regulatory regions was performed using i-*cis*Target [42,43].

## RNA-sequencing analysis

For transcriptomic analysis, total RNA was extracted from 8 to 15 abdominal cuticles from male flies 6 days after eclosion using the Promega ReliaPrep RNA Tissue Miniprep kit (z6112, Promega, U.S.A.). Three biological replicates were prepared for each genotype. RNA samples were sent to Kazusa Genome Technologies (Kisarazu, Japan) for 3′ mRNA-seq analysis. A complementary DNA library was prepared using the QuantSeq 3′ mRNA-Seq Library Prep kit for Illumina (FWD) (015.384, Lexogen). Sequencing was conducted using Illumina NextSeq 500 and NextSeq 500/550 High Output Kit v.2.5 (75 cycles) (20024906, Illumina). Raw reads were analyzed using the BlueBee Platform (Lexogen) for trimming, alignment to the

*Drosophila* genome, and read counting. Count data were analyzed using the Wald test with DESeq2. The results were deposited in the DNA Data Bank of Japan (DDBJ) under accession no. DRA014922. The Gene Ontology enrichment analysis was performed using the DAVID bioinformatics resource [76]. The analysis of putative transcription factor-binding sites in the gene regulatory regions was performed using i-*cis*Target [42,43].

## Molecular cloning

The primers used for molecular cloning are listed in the S6 Table. For the *pMT-catalytic domain of SP*, the coding sequences of the catalytic domains of hemolymph SPs [49] and the Easter signal peptide were PCR-amplified from cDNA.

For *pMT-Spz-3xFLAG*, the coding sequence of Spz was PCR-amplified from the genomic DNA of *UASp-Spz-HA* flies, kindly provided by Dr. D. Stein, [77], and the coding sequence of 3xFLAG was PCR-amplified from the *pUAST-attB/Flag-HA[STOP]* vector, kindly provided by Dr. T. Chihara. For *pMT-Spz$^{uc}$-3xFLAG*, the "CGC-GTT (Arg-Val)" to "CTT-AAT (Leu-Asn)" mutation was introduced via PCR-based site-directed mutagenesis from *pMT-Spz-3xFLAG*.

For *pMT-pro-Psh$^{WT}$*, *pMT-pro-Hayan (*isoform A*)*, *pMT-pro-SPE*, and *pMT-pro-ηTrypsin*, the coding sequences of each SP were PCR-amplified from cDNA.

For *pMT-pro-Psh$^{S339A}$*, the S339A (TCC to GCC) catalytic-dead mutation [32] was introduced via PCR-based site-directed mutagenesis from *pMT-pro-Psh$^{WT}$*.

For *pMT-pro-Psh$^{H143E}$*, the H143E (CAC to GAG) cleavage-resistant mutation [32] was introduced via PCR-based site-directed mutagenesis from *pMT-pro-Psh$^{WT}$*.

For *pMT-Spz-3xFLAG-P2A-pro-Psh$^{WT}$* and *pMT-Spz$^{uc}$-3xFLAG-P2A-pro-Psh$^{WT}$*, the coding sequences of Spz-3xFLAG or Spz$^{uc}$-3xFLAG were PCR-amplified from *pMT-Spz-3xFLAG* or *pMT-Spz$^{uc}$-3xFLAG* plasmids, respectively, with GGGSGGG linker and P2A sequences at the 3' terminus [51,52]. The coding sequence of pro-Psh$^{WT}$ was PCR-amplified from *pMT-pro-Psh$^{WT}$*.

For *pMT-Spz-3xFLAG-P2A-pro-Psh$^{S339A, or H143E}$*, the coding sequence of Spz-3xFLAG-P2A was PCR-amplified from *pMT-Spz-3xFLAG-P2A-pro-Psh$^{WT}$*. The pro-Psh$^{S339A}$ and pro-Psh$^{H143E}$ was PCR-amplified from *pMT-pro-Psh$^{S339A}$* and *pMT-pro-Psh$^{H143E}$*, respectively.

For *pMT-Spz-3xFLAG-P2A-pro-Psh$^{H143G or H143A}$*, the coding sequence of Spz-3xFLAG-P2A was PCR-amplified from *pMT-Spz-3xFLAG-P2A-pro-Psh$^{WT}$*. Either the H143G (CAC to GGT) or H143A (CAC to GCA) mutation was introduced via PCR-based site-directed mutagenesis from *pMT-Spz-3xFLAG-P2A-pro-Psh$^{WT}$*.

For *pMT-Spz-3xFLAG-P2A-pro-Psh$^{\Delta136-141, \Delta142-147, \Delta148-153,}$* and *$^{\Delta136-153}$*, the coding sequence of *Spz-3xFLAG-P2A* was PCR-amplified from *pMT-Spz-3xFLAG-P2A-pro-Psh$^{WT}$*. The coding sequence of pro-Psh carrying each deletion was PCR-amplified from *pMT-Spz-3xFLAG-P2A-pro-Psh$^{WT}$*.

All amplicons were ligated into the EcoRI/XhoI-digested *pMT/V5-HisA* vector (Thermo Fisher Scientific, U.S.A.) using the NEBuilder HiFi DNA Assembly (E2621L, NEB, U.S.A.).

## Cell culture

To test the ability of hemolymph SPs to cleave Spz, a total of $1.5 \times 10^6$ S2 cells were cultured in 6-well plates at 25°C overnight in Schneider's medium (21720001, Thermo Fisher Scientific, U.S.A.) supplemented with 10% (v/v) heat-incubated fetal bovine serum (FBS), 100 U/mL penicillin, and 100 μg/mL streptomycin (168–23191, FUJIFILM Wako, Japan). Then, the S2 cells were transfected with 400 ng of the *pMT/V5-HisA* construct bearing the catalytic domain of each SP in Schneider's medium supplemented with 5% (v/v) heat-incubated FBS, 100 U/mL

penicillin, and 100 μg/mL streptomycin using the Effectene Transfection Reagent (301427, QIAGEN, Germany). After incubation for 24 h, the medium was replaced with fresh medium, and 500 μM CuSO$_4$ was used to induce each SP. After incubation for 24 h, cells were washed with 1 mL PBS and lysed with 200 μL lysis buffer (1.0% NP-40, 50 mM Tris-HCl (pH8.0), 150 mM sodium chloride, 0.5% sodium deoxycholate, and 0.1% sodium dodecyl sulfate) supplemented with 1 × cOmplete ULTRA EDTA-free protease inhibitor cocktail (5892953001, Roche, Switzerland). The samples were centrifuged at 14,000 × $g$ for 10 min at 4˚C, and the supernatants were collected. The supernatants were mixed with 6 × Laemmli sample buffer to a final concentration of 1 × and boiled at 98˚C for 5 min. The protein concentration was measured using the BCA Protein Assay Kit.

To test whether treatment of S2 cells with H$_2$O$_2$, cycloheximide, or TritonX-100 activates pro-Psh and -Hayan and leads to Spz cleavage, a total of $6.0 \times 10^5$ S2 cells were cultured in 6-well plates at 25˚C overnight in Schneider's medium supplemented with 10% (v/v) heat-incubated FBS, 100 U/mL penicillin, and 100 μg/mL streptomycin. Then, the S2 cells were transfected with 400 ng of the desired *pMT/V5-HisA* constructs in Schneider's medium supplemented with 5% (v/v) heat-incubated FBS, 100 U/mL penicillin, and 100 μg/mL streptomycin using the Effectene Transfection Reagent. After incubation for 24 h, the medium was replaced with Schneider's medium supplemented with 10% (v/v) heat-incubated FBS, 100 U/mL penicillin, and 100 μg/mL streptomycin, and 500 μM CuSO$_4$ was used to induce protein expression. After incubation for 24 h, each chemical agent was added to the medium as follows: H$_2$O$_2$ (080–01186, FUJIFILM Wako, Japan) at 15 mM, cycloheximide (c7698, Sigma-Aldrich, U.S.A.) at 2.5 or 5.0 μg/mL, and TritonX-100 (35501–15, Nacalai tesque, Japan) at 0.005 or 0.01%. After incubation for 18 h, the medium was changed to a new Schneider's medium, and Hoechst 33342 (H3570, Thermo Fisher Scientific) was added at 0.8 μM and Propidium Iodide (P4170, Sigma-Aldrich) was added at 5 μg/mL. After incubation for 20 min, images were captured using a Leica DMi8-AFC microscope (Leica Microsystems, Germany). After capturing images, cells were washed with 1 mL PBS and lysed with 100–150 μL lysis buffer supplemented with 1 × cOmplete ULTRA EDTA-free protease inhibitor cocktail. The samples were centrifuged at 14,000 × $g$ for 10 min at 4˚C, and the supernatants were collected. The supernatants were mixed with 6 × Laemmli sample buffer at a final concentration of 1 × and boiled at 98˚C for 5 min. The protein concentration was measured using the BCA Protein Assay Kit.

For *MP1* and *26-29-p* knockdown in S2 cells, double-stranded RNAs (dsRNAs) were generated using the T7 RiboMAX Express RNAi System (P1700, Promega, U.S.A.) according to the manufacturer's instructions. For the negative control dsRNA, pBlueScript II SK (+) was used as a template, while for dsRNA targeting *MP1*, fly genomic DNA was used as a template, and for dsRNA targeting *26-29-p*, larval cDNA reverse-transcribed from total RNA was used as a template. The following primers were used to generate each dsRNA: negative control dsRNA, 5'-TAATACGACTCACTATAGGTAAATTGTAAGCGTTAATATTTTG-3' and 5'-TAATAC GACTCACTATAGGAATTCGATATCAAGCTTATCGAT-3'; MP1#1 dsRNA, 5'-TAATAC GACTCACTATAGGCCACTTTAGCGTCCCAGCAC-3' and 5'-TAATACGACTCACTATA GGCCGTAGGAGACCACACCAGC-3'; MP1#2 dsRNA, 5'-TAATACGACTCACTATAGGA CAGTCGGTAAAAACGGACG-3' and 5'-TAATACGACTCACTATAGGGGCGTATCTTT GGTTGCATT-3'; 26-29-p#1 dsRNA, 5'-TAATACGACTCACTATAGGAGTCGCCGCATT ATCCCTCCAG-3' and 5'-TAATACGACTCACTATAGGGCTCGTGTTCCGTGTCGCTG TG-3'; and 26-29-p#2 dsRNA, 5'-TAATACGACTCACTATAGGAGGGGCTATAACACCCT GCTGGG-3' and 5'-TAATACGACTCACTATAGGCGTGCCTTCAACTCCTCTTCG-3'. For the experiments shown in Fig 3E and 3F, a total of $1.0 \times 10^6$ S2 cells were cultured in 6-well plates at 25˚C overnight in Schneider's medium supplemented with 10% (v/v) heat-incubated

FBS, 100 U/mL penicillin, and 100 μg/mL streptomycin. Then, the S2 cells were co-transfected with 380 ng *pMT-catalytic domain of the Psh/Hayan* and 20 ng *pAc5co-Hygro* ([78]; kindly provided by Dr. S. Nagata) using Effectene Transfection Reagent and selected in the presence of hygromycin (084–07681, FUJIFILM Wako, Japan). Stably transfected S2 cells were then seeded in 24-well plates and treated with dsRNA to a final concentration of 10 μg/mL every 2–3 days for at least 2 weeks. The expression of active Psh and Hayan was induced by 500 μM $CuSO_4$ for 24 h. For protein preparation, cells were lysed with 100 μL lysis buffer supplemented with 1 × cOmplete ULTRA EDTA-free protease inhibitor cocktail. The samples were centrifuged at $14,000 \times g$ for 10 min at 4˚C, and the supernatants were collected. The supernatants were mixed with 6 × Laemmli sample buffer at a final concentration of 1 × and boiled at 98˚C for 5 min. The protein concentration was measured using the BCA Protein Assay Kit.

Total RNA was extracted using the Promega ReliaPrep RNA Tissue Miniprep System (Z6110, Promega, U.S.A.). cDNA was generated from 400 ng of total RNA using PrimeScript RT Master Mix (Perfect Real Time) (RR036A, Takara Bio, Japan). For the experiments shown in Figs 5B and 6B, a total of $6.0 \times 10^5$ dsRNA-treated S2 cells were cultured in 6-well plates at 25˚C overnight in Schneider's medium supplemented with 10% (v/v) heat-incubated FBS, 100 U/mL penicillin, and 100 μg/mL streptomycin. S2 cells were co-transfected with 400 ng of *pMT-Spz-3xFLAG*, and *pMT-pro-Psh$^{WT}$* (Fig 5B) or *pMT-Spz-3xFLAG-P2A-pro-Psh$^{WT}$* (Fig 6B) constructs in Schneider's medium supplemented with 5% (v/v) heat-incubated FBS, 100 U/mL penicillin, and 100 μg/mL streptomycin using the Effectene Transfection Reagent. After incubation for 24 h, the medium was replaced with fresh Schneider's medium supplemented with 10% (v/v) heat-incubated FBS, 100 U/mL penicillin, and 100 μg/mL streptomycin, and 500 μM $CuSO_4$ was used to induce protein expression. After incubation for 24 h, 15 mM of $H_2O_2$ was added to the medium. After incubation for 18 h, the cells were washed with 1 mL PBS and lysed with 50 μL lysis buffer supplemented with 1 × cOmplete ULTRA EDTA-free protease inhibitor cocktail. The samples were centrifuged at $14,000 \times g$ for 10 min at 4˚C, and the supernatants were collected. The supernatants were mixed with 6 × Laemmli sample buffer at a final concentration of 1 × and boiled at 98˚C for 5 min. The protein concentration was measured using the BCA Protein Assay Kit. For the experiment shown in Fig 6A, S2 cells were seeded in 24-well plates and treated with dsRNA to a final concentration of 10 μg/mL every 2–3 days for at least 2 weeks. Total RNA was extracted using the Promega ReliaPrep RNA Tissue Miniprep System. cDNA was generated from 400 ng of total RNA using the PrimeScript RT Master Mix (Perfect Real Time).

### Wing observation

Adult male flies 24–30 h after eclosion were injected with 2.5 mM dihydroethidium (D11347, Thermo Fisher Scientific, U.S.A.) and 4 mM Hoechst 33342 in the thoraces using FemtoJet 4i (Eppendorf). After incubation for 1 h, the wings were removed from the flies and mounted using Euparal (35358–86, Nacalai tesque, Japan). Images were captured using the Leica DMi8-AFC microscope (Leica Microsystems, Germany).

### Protein preparation from *Drosophila* samples for western blotting

For the experiment shown in Fig 1F, five adult male flies were lysed with 100 μL of lysis buffer supplemented with 1 × cOmplete ULTRA EDTA-free protease inhibitor cocktail. Samples were homogenized and centrifuged at 15,000 rpm for 15 min at 4˚C, and the supernatants were collected. The supernatants were mixed with 6 × Laemmli sample buffer to a final concentration of 1 × and boiled at 98˚C for 5 min. The protein concentration was measured using the BCA Protein Assay Kit.

For the experiment shown in Fig 2D, adult *Drosophila* hemolymph was collected following the method described in a previous study [79] with modifications. Briefly, female flies (20–25) were anesthetized on a $CO_2$ pad and pricked in the thorax using a metal needle. The treated flies were transferred to a 0.5-mL centrifuge tube pierced with a 25G needle at the bottom. The 0.5-mL tube was transferred to a 1.5 mL centrifuge tube containing 15 μL of PBS with $1 \times$ cOmplete ULTRA EDTA-free protease inhibitor cocktail. The assembled tubes were centrifuged at 5,000 rpm for 5 min at 4˚C. The 0.5 mL centrifuge tube was removed, and the 1.5 mL centrifuge tube now containing hemolymph was centrifuged a second time at 5,000 rpm for 5 min at 4˚C. The supernatants were transferred to a 1.5 mL centrifuge tube, and the 1.5 mL tube was centrifuged at 14,000 *g* for 15 min at 4˚C. The supernatants were collected in another tube, mixed with $6 \times$ Laemmli sample buffer at a final concentration of $1 \times$, and boiled at 98˚C for 5 min. The protein concentration was measured using the BCA Protein Assay Kit.

For the experiment shown in Fig 3D, adult male flies were dissected in PBS, and the abdominal cuticles were collected by removing the intestines and male reproductive organs. Five abdominal cuticles were lysed with 150 μL of lysis buffer supplemented with $1 \times$ cOmplete ULTRA EDTA-free protease inhibitor cocktail. Samples were homogenized and centrifuged at $14,000 \times g$ for 3 min at 4˚C, and the supernatants were collected. The supernatants were mixed with $6 \times$ Laemmli sample buffer at a final concentration of $1 \times$ and boiled at 98˚C for 5 min. The protein concentration was measured using the BCA Protein Assay Kit.

### Generation of antibodies

Anti-SPE antibodies were generated in Eurofins Genomics K.K. (Tokyo, Japan). The C-terminus of the SPE protein, T387-P400 (TGAFIDWIKQKLEP), was synthesized with cysteine at the N-terminus to conjugate with a carrier protein, keyhole limpet hemocyanin. The carrier-peptide conjugate was injected into rabbits, and the whole antiserum was collected.

To generate an anti-Bbd antibody, the coding sequence of the Bbd protein was PCR-amplified without a signal sequence. The following primers were used: forward primer, 5'-CGCGG ATCCAATATACAGCGAAATGAGGAC-3', and reverse primer, 5'-CCCAAGCTTCTAATA GAAAATATTTCC-3'. The amplicon was ligated to BamHI/HindIII-digested *pCold I* vector (3361, Takara Bio, Japan) to generate *pCold-Bbd*. The *pCold-Bbd* was introduced into *Escherichia coli* BL21, and isopropyl β-D-1-thiogalactopyranoside (IPTG) was added to a final concentration of 1 mM to express the truncated Bbd protein. After incubation for 24 h at 15˚C, the sample was centrifuged at $6,000 \times g$ for 2 min at 4˚C. From the obtained pellet, the truncated Bbd protein was purified with a His-tag using Ni-NTA agarose. The purified protein was sent to Hokudo Co., Ltd (Sapporo, Japan) and injected into a rabbit as an antigen to obtain antiserum.

### Western blotting

Each sample was separated using sodium dodecyl-sulfate polyacrylamide gel electrophoresis (SDS-PAGE). The proteins were then transferred to Immobilon-P polyvinylidene fluoride (PVDF) membranes (IPVH00010, Millipore, U.S.A.) for immunoblotting. Membranes were blocked with 5% skim milk diluted in 1× TBST. Immunoblotting was performed using the antibodies mentioned below, which were diluted in 5% skim milk. The signals were visualized using the Immobilon Western Chemiluminescent HRP Substrate (WBKLS0500; Millipore, U. S.A.) and FUSION SOLO. 7S. EDGE (Vilber-Lourmat, France).

The primary antibodies used were rabbit anti-Bbd polyclonal antibody (1:500, generated in this study), rat anti-Spz C106 polyclonal antibody (1:1000, kindly provided by Dr. M. Yamamoto-Hino and Dr. S. Goto, [80]), mouse anti-V5 monoclonal antibody (1:3,000, R960-25,

Thermo Fisher Scientific), rabbit anti-SPE polyclonal antibody (1:1000, generated in this study), and mouse anti-FLAG M2 monoclonal antibody (1:2000, F1804, Sigma-Aldrich). In addition, mouse anti-alpha-tubulin (DM1A) monoclonal antibody (1:2000, T9026, Sigma-Aldrich) was used as a loading control. The secondary antibodies used were HRP-linked goat anti-rabbit IgG (1:5,000, 7074S, Cell Signaling Technology), HRP-linked goat anti-rat IgG (1:5,000, 7077S, Cell Signaling Technology), and HRP-conjugated goat/rabbit/donkey anti-mouse IgG (1:5000, W402B, Promega).

## Quantitative RT-PCR

Total RNA was extracted from five adult male flies (three adult male flies, in S8B–S8D Fig), 4–6 male abdominal cuticles, and 60 wings using the Promega ReliaPrep RNA Tissue Mini-prep system. cDNA was generated from 400 ng (male whole-body cell lysate) or 200 ng (male abdominal cuticle) of total RNA using PrimeScript RT Master Mix (Perfect Real Time). Quantitative PCR was performed using the TB Green Premix Ex Taq (Tli RNaseH Plus) (RR820W, Takara Bio, Japan) and a Quantstudio6 Flex Real-Time PCR system (Thermo Fisher Scientific, U.S.A.). The experiments shown in Fig 4A–4F were repeated independently three times (with six samples and 30 flies). Each point in the Fig indicates the average value of six samples in a single experiment. For other quantitative RT-PCR experiments, every point in the Fig represents the value of a single sample. The primers used are listed in the S7 Table.

## *In vivo* RNAi screening for *Drosophila* SP genes

Each UAS-RNAi line was crossed with *WP-QF2*, *QUAS-Dark$^{sh}$*, and *Da-Gal4* flies. A sample batch containing 4–6 adult male flies (6–7 days after adult eclosion) was homogenized with a Multi-Beads Shocker (Yasui Kikai, Japan) set to 1500 rpm for 15 s. The homogenate was used for RNA extraction with TRIzol (Thermo Fisher Scientific, U.S.A.), using chloroform, 2-propanol, and ethanol, according to the manufacturer's instructions. Total RNA was dissolved in 25 μL diethylpyrocarbonate-processed water. cDNA was synthesized with a TaKaRa Prime-Script RT reagent Kit with a gDNA Eraser using 500–600 ng of total RNA, obtaining 20 μL of cDNA solution. qRT-PCR was performed using TaKaRa Premix Ex Taq (Tli RNaseH Plus) on a LightCycler 480 system (Roche Applied Science, Germany). The reaction system was composed as follows: 5 μL of a five-fold-diluted sample of obtained cDNA, 10 μL TaKaRa Premix Ex Taq, 0.8 μL of primer pair mixture (10 μM each), and 8.2 μL of Milli-Q water. The PCR cycle was as follows: denaturing at 95˚C for 30 s → PCR cycle at 95˚C for 5 s then at 55˚C for 25 s for 45–60 cycles → melting curve at 95˚C for 10 s. *Rpl32* or *RNApolII* was used as the internal control. For each data set, the value of the wild-type sample was used as a calibrator, and the relative ratios of the other genotypes were calculated accordingly.

## Statistical analysis

Statistical analyses were performed using GraphPad Prism version 8. Two-tailed Welch's t tests were used to test between samples. One-way ANOVA with Tukey's multiple comparison test was used to test among groups. One-way ANOVA with Dunnett's multiple comparison test was used to compare the control and multiple samples. All experiments were performed at least twice to confirm the reproducibility (unless otherwise indicated). Statistical significance is shown as follows; ns: $P > 0.05$; *: $P < 0.05$; **: $P < 0.01$; ***: $P < 0.001$; ****: $P < 0.0001$.

## Supporting information

**S1 Fig. Apoptosis deficiency of wing epidermal cells induces Toll activation.** (A–C) Quantitative RT-PCR of *Drs* (A), *BomS1* (B), and *Bbd* (C) in the whole body of control male flies ($WP^{QF}>+$), apoptosis-deficient male flies ($WP^{QF}>Dark^{sh}$), and male flies with septic injury with *M. luteus* ($WP^{QF}>+$, *M. luteus*) raised on a standard diet. Expression levels of Toll target genes under each condition are listed in the table, with those of flies with *M. luteus* infection as 100%. n = 6. Data are mean with SEM. Each dot represents a replicate. Statistical analysis was performed using one-way ANOVA with Tukey's multiple comparison test. *: $P < 0.05$; **: $P < 0.01$; ***: $P < 0.001$; ****: $P < 0.0001$.
(TIF)

**S2 Fig. Toll-mediated effector proteins are increased in the hemolymph of apoptosis-deficient flies.** Proteomic analysis of adult hemolymph from control ($WP^{QF}>+$) and apoptosis-deficient ($WP^{QF}>Dark^{sh}$) male flies at 6 days after eclosion. Three biological replicates were prepared for each genotype. List of upregulated proteins ($\log_2$Fold change $\geq 1.0$, adjusted $P$ value $< 0.1$) having the putative signal sequence in the hemolymph of apoptosis-deficient flies compared with that of control flies. A color scale on the left side of the list shows whether the genes encoding each protein are known to be an immune responsive core gene (yellow), have the putative Dorsal-related immune factor (Dif) binding sites (green), or have the putative signal sequence (light blue).
(TIF)

**S3 Fig. Differentially expressed genes in apoptosis-deficient flies, whose expression depends on Spz cleavage.** From the RNA-seq analysis, the differentially expressed genes, whose expression was upregulated in apoptosis-deficient flies ($WP^{QF}>Dark^{sh}$ versus $WP^{QF}>+$) but downregulated in $spz^{uc}$-background flies ($WP^{QF}>Dark^{sh}$, $spz^{uc}$ versus $WP^{QF}>Dark^{sh}$), were listed with Benjamini–Hochberg adjusted $P$ value $<0.05$ after Wald test. i-*cis*Target analysis identified 28 putative Dif target genes in the 68 genes, indicating that apoptosis deficiency induces Toll pathway activation through Spz cleavage.
(TIF)

**S4 Fig. Expression of SP genes in whole-body fly and S2 cell samples.** (A–E) Quantitative RT-PCR of *SPE* (A), *MP1* (B), *easter* (C), *Hayan* (D), and *psh* (E) in the whole body of control ($w^{iso31}$) male fly and S2 cell samples. n = 4. Data are mean with SEM. Each dot represents a replicate. Statistical analysis was performed using two-tailed Welch's t test. ns: $P > 0.05$; **: $P < 0.01$; ***: $P < 0.001$; ****: $P < 0.0001$.
(TIF)

**S5 Fig. *SPE, MP1* double mutants show stronger impairment of Toll activation upon infection with *M. luteus* than *SPE* single mutants.** (A–C) Quantitative RT-PCR of *Drs* (A), *BomS1* (B), and *Bbd* (C) in the whole body of control ($w^{iso31}$) and SP mutant ($SPE^{SK6}$, $MP1^{SK6}$, or $SPE^{SK6} + MP1^{SK6}$) male flies raised on a standard diet. Each group of flies was either untreated or treated, with either clean or septic injury. Fly samples were collected 24 hours after treatment. n = 6. Data are mean with SEM. Each dot represents a replicate. Statistical analysis was performed using one-way ANOVA with Tukey's multiple comparison test. ns: $P > 0.05$; ****: $P < 0.0001$.
(PDF)

**S6 Fig. Upstream SP, modSP, does not play a major role in Toll pathway activation in apoptosis-deficient flies.** (A-C) Quantitative RT-PCR of *Drs* (A), *BomS1* (B), and *Bbd* (C) in the abdominal cuticles of control ($WP^{QF}>+$), apoptosis-deficient ($WP^{QF}>Dark^{sh}$), or apoptosis-

deficient with *modSP$^1$* background male flies raised on an antibiotics-supplemented diet.
n = 6. Data are mean with SEM. Each dot represents a replicate.
(TIF)

**S7 Fig. H$_2$O$_2$ treatment of S2 cells induces Spz cleavage depending on overexpressed pro-Psh.** (A, B, and E) Western blotting of S2 cell lysates against FLAG-tagged Spz (anti-FLAG antibody) and V5-tagged pro-Hayan (A) or pro-Psh (B and E) (anti-V5 antibody). (A) S2 cells were transfected with either *pMT-Spz-3xFLAG* or *pMT-Spz$^{uc}$-3xFLAG* and *pMT-pro-Hayan-PA-V5*. H$_2$O$_2$ treatment of 15 mM for 18 h. (B) S2 cells were transfected with either *pMT-Spz-3xFLAG* or *pMT-Spz$^{uc}$-3xFLAG* and *pMT-pro-Psh$^{WT}$-V5*. H$_2$O$_2$ treatment of 15 mM for 18 h. The signal ratio of active Spz (~20 kDa)/pro-Spz (~37 kDa) was approximately 4-fold greater in the H$_2$O$_2$ treated condition, calculated using FUSION SOLO. 7S. EDGE software (Vilber-Lourmat). (C) Western blotting of S2 cell lysates against Spz (anti-Spz C106 antibody) and V5-tagged pro-Psh (anti-V5 antibody). S2 cells were transfected with *pMT-pro-Psh$^{WT}$-V5*. H$_2$O$_2$ treatment of 15 mM for 18 h. (D) Schematic representation of the plasmid tandemly bearing the coding sequences of Spz-3xFLAG and pro-Psh-V5-6xHis with the GGGSGGG linker and viral P2A sequence between them. After transcription, a single mRNA is translated with 2A-mediated hydrolysis, independently resulting in the production of both Spz and pro-Psh protein. (E) S2 cells were transfected with *pMT-Spz-3xFLAG-P2A-pro-Psh$^{WT or S339A}$-V5*. H$_2$O$_2$ treatment of 15 mM for 18 h.
(PDF)

**S8 Fig. Overexpression of pro-Hayan and pro-Psh activates the Toll pathway *in vivo*.** (A) Representative images of adult male flies overexpressing *LacZ*, *pro-Hayan*, *pro-Hayan mutant*, or *pro-Psh* by *Lpp-Gal4, tub-Gal80$^{ts}$* (*Lpp$^{ts}$*). Flies were reared at 18°C until 2–3 days after eclosion, then reared at 29°C for 48 h. An arrowhead indicates a melanotic mass. (B–D) Quantitative RT-PCR of *Drs* (B), *BomS1* (C), and *Bbd* (D) in the whole body of male flies overexpressing *LacZ*, *pro-Hayan*, *pro-Hayan mutant*, or *pro-Psh* by *Lpp-Gal4, tub-Gal80$^{ts}$* (*Lpp$^{ts}$*). n = 3. Data are mean with SEM. Each dot represents a replicate. Statistical analysis was performed using one-way ANOVA with Tukey's multiple comparison test. **: $P < 0.01$; ***: $P < 0.001$; ****: $P < 0.0001$.
(TIF)

**S9 Fig. H$_2$O$_2$ treatment induces morphological apoptotic changes in S2 cells.** Representative images of S2 cells with Hoechst and PI staining. S2 cells were transfected with *pMT-Spz-3xFLAG* and *pMT-pro-Psh$^{WT}$-V5*. H$_2$O$_2$ treatment of 15 mM, TritonX-100 treatment of 0.005 or 0.01%, or CHX treatment of 2.5 or 5.0 μg/mL for 18 h. H$_2$O$_2$-treated S2 cells showed apoptotic traits, such as membrane blebbing (arrowheads) and nucleus condensation (Hoechst stain). TritonX-treated cells showed the increase in cell membrane permeability (PI). CHX-treated cells showed apoptotic traits, such as apoptotic body formation and nucleus condensation (Hoechst stain). Scale bars: 100 μm.
(PDF)

**S10 Fig. Psh is activated upon H$_2$O$_2$ treatment in a different manner from its activation by 26-29-p.** (A) Quantitative RT-PCR of *26-29-p* in the whole body of control (*w$^{iso31}$*) male fly and S2 cell samples. n = 4. Data are mean with SEM. Each dot represents a replicate. Statistical analysis was performed using two-tailed Welch's t test. ***: $P < 0.001$. (B and C) Western blotting of S2 cell lysates against FLAG-tagged Spz (anti-FLAG antibody) and V5-tagged pro-Psh (anti-V5 antibody). (B) S2 cells were transfected with *pMT-Spz-3xFLAG* and *pMT-pro-Psh$^{WT or H143E}$-V5*. H$_2$O$_2$ treatment of 15 mM for 18 h. (C) S2 cells were transfected with *pMT-Spz-3xFLAG-P2A-pro-Psh$^{WT, Δ136–141, Δ142–147, Δ148–153, or Δ136-153}$-V5*. H$_2$O$_2$ treatment of 15 mM

for 18 h.
(TIF)

**S11 Fig. Gene expressions of *spz*, *Hayan*, and *psh*, but not *Hemese*, are detected in wings of apoptosis-deficient flies.** (A–D) Quantitative RT-PCR of *spz* (A), *Hayan* (B), *psh* (C), and *Hemese* (D) in the whole body of control ($w^{iso31}$) male fly, S2 cell samples, and wings of apoptosis-deficient male flies 0 or 3 days after eclosion. n = 3. Data are mean with SEM. Each dot represents a replicate. Statistical analysis was performed using one-way ANOVA with Tukey's multiple comparison test. *: $P < 0.05$; **: $P < 0.01$; ***: $P < 0.001$; ****: $P < 0.0001$.
(TIF)

**S12 Fig. Spz cleavage is observed by mixing S2 cell lysate containing Spz with lysate containing a *Drosophila* trypsin, ηTrypsin.** (A-C) Western blotting of S2 cell lysates against V5-tagged serine proteases (anti-V5 antibody) (A), Spz (anti-Spz C106 antibody) (B), and V5-tagged ηTrypsin and Spz (anti-V5 antibody) (C). (A and B) S2 cells overexpressed pro-SPE, pro-ηTrypsin, the catalytic domain of SPE, or the catalytic domain of ηTrypsin. Spz cleavage was observed when pro-ηTrypsin or the catalytic domain of SPE was overexpressed. (C) Lysates of S2 cells overexpressing pro-ηTrypsin, Spz$^{WT}$, and Spz$^{uc}$ were collected, and Spz-containing lysates were mixed with pro-ηTrypsin-containing lysates. Incubation at 25˚C for 1, 6, or 18 h. When Spz$^{WT}$ was incubated with pro-ηTrypsin, active Spz was observed 6 and 18 h after incubation, suggesting that *Drosophila* ηTrypsin can cleave Spz in a similar manner to mammalian trypsins. Notably, while pro-ηTrypsin is catalytically active, the putative active form ηTrypsin does not show catalytic activity against Spz, different from other standard serine proteases that need to be cleaved for their activation. This uniqueness of ηTrypsin is potentially due to the shortness of the pro-domain of ηTrypsin (the pro-domain of SPE is composed of 107 amino acids, while that of ηTrypsin is composed of 5); thus, the putative pro-domain of ηTrypsin is also required for the active conformation of ηTrypsin.
(TIF)

**S13 Fig. Hypothetical model of damage-dependent activation of the Toll pathway in apoptosis-deficient flies.** Hayan is auto-activated upon induction of its expression and Psh is activated by $H_2O_2$ in the secretory pathway of necrotic wing epithelia. Activated Hayan and Psh activate unknown SPs, and then pro-Spz is cleaved in the secretory pathway.
(TIF)

**S1 Table. GO terms enriched in 68 differentially expressed genes from the RNA-seq analysis.**
(XLSX)

**S2 Table. In vivo RNAi screening for *Drosophila* serine protease genes.** qRT-PCR analysis of *Drs* (n = 3) and percentage of individuals with melanized wings (n > 20) for each *UAS-RNAi* lines crossed with *WP-QF2*, *QUAS-Dark*$^{sh}$, *Da-Gal4* fly. The value of '*Drs* induction' was relative ratio to control line (*WP-QF2*, *QUAS-Dark*$^{sh}$, *Da-Gal4*, *lacZ-IR*).
(XLSX)

**S3 Table. LC settings for proteome analysis.**
(XLSX)

**S4 Table. MS settings for proteome analysis.**
(XLSX)

**S5 Table. Proteome Discoverer 2.2 settings for proteome analysis.**
(XLSX)

**S6 Table. Primers used for creating pMT plasmids.**
(XLSX)

**S7 Table. Primers used for quantitative RT-PCR.**
(XLSX)

## Acknowledgments

We thank M. Yamamoto-Hino, S. Goto, B. Lemaitre, M. Brankatschk, S. Eaton WJ. Lee, JM. Reichhart, D. Stein, the Bloomington *Drosophila* stock center, and Vienna *Drosophila* Resource Center for fly stocks; N. Perrimon, T. Chihara, and S. Nagata for providing the plasmid; M. Yamamoto-Hino and S. Goto for providing the antibody; and S. Hirayama, S. Murata, and One-stop Sharing Facility Center for Future Drug Discovery for LC-MS/MS analysis. We thank members of the M.M. laboratory for their technical assistance and discussions, in particular, K. Takenaga for the preparation of the fly food, Y. Fujioka for collecting hemolymph, T. Katsuyama for the generation of the pBac[3xP3-DsRed_polyA_Scarless_TK] plasmid and assistance with the generation of the *spz$^{uc}$* knockin allele by CRISPR/Cas9, and N. Shinoda for the support in the LC-MS/MS analysis. We deeply appreciate the encouragement given by S. Iwanaga.

## Author Contributions

**Conceptualization:** Shotaro Nakano, Masayuki Miura.

**Data curation:** Shotaro Nakano.

**Formal analysis:** Shotaro Nakano.

**Funding acquisition:** Masayuki Miura.

**Investigation:** Shotaro Nakano, Kei Nishimura, Asuka Takeishi, Yoshio Yamauchi.

**Methodology:** Shotaro Nakano, Hina Kosakamoto.

**Resources:** Shotaro Nakano.

**Supervision:** Soshiro Kashio, Fumiaki Obata, Erina Kuranaga, Takahiro Chihara, Toshiaki Isobe, Masayuki Miura.

**Validation:** Shotaro Nakano.

**Visualization:** Shotaro Nakano.

**Writing – original draft:** Shotaro Nakano, Masayuki Miura.

**Writing – review & editing:** Soshiro Kashio, Kei Nishimura, Asuka Takeishi, Hina Kosakamoto, Fumiaki Obata, Erina Kuranaga, Takahiro Chihara.

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
