## [Decision Letter · Decision Letter 0]

21 Dec 2022

Dear Dr Miura,

Thank you very much for submitting your Research Article entitled 'Damage sensing mediated by serine proteases Hayan and Persephone for Toll pathway activation in apoptosis-deficient flies' to PLOS Genetics.

The manuscript was fully evaluated at the editorial level and by independent peer reviewers. The reviewers appreciated the attention to an important problem, but raised some substantial concerns about the current manuscript. Based on the reviews, we will not be able to accept this version of the manuscript, but we would be willing to review a much-revised version. We cannot, of course, promise publication at that time.

If you decide to revise the manuscript for further consideration at PLOS Genetics, please aim to resubmit within the next 60 days, unless it will take extra time to address the concerns of the reviewers, in which case we would appreciate an expected resubmission date by email to plosgenetics@plos.org.

We are sorry that we cannot be more positive about your manuscript at this stage. Please do not hesitate to contact us if you have any concerns or questions.

Yours sincerely,

Norbert Perrimon

Academic Editor

PLOS Genetics

Gregory P. Copenhaver

Editor-in-Chief

PLOS Genetics

Reviewer's Responses to Questions

**Comments to the Authors:**

Reviewer #1: This is a very interesting study that analyzes the serine protease pathway leading to Toll activation in a Drosophila model of sterile stimulation. Although, many connections have been previously shown by other studies, this article links many disparate observations and clarify some important points combining both in vivo and cell culture approaches. Thus, this article is an important addition to the field of Drosophila immunity and wound healing.

I have however some recommendations and suggestions to improve the impact of this article.

Major recommendations

1) Figure 1A: it would have been interesting to know what is the level of Drs expression in the Dark background compared not only to unchallenged flies but also to flies infected with a Gram-positive bacterium (M. luteus). Just to have an idea of the range of activation of the Toll pathway. The authors could simply write the level of Drs expression in apoptosis deficient background represent x% of the level of flies infected with a G+. This will allow the community to estimate the level of Toll pathway activation in their system.

2) The claims that SPE and MP1 redundantly contribute to Spz cleavage should be tested by using a SPE, MP1 double mutants. We should to know if spe,MP1 double mutants display a stronger impairment of Toll pathway activation than SPE or MP1 single mutants in their model of sterile inflammation and upon septic injury. If this is too challenging to combine, MP1, SPE with the genetic elements inducing apoptosis, the authors could inject ROS.

Suggestions to improve the paper

I have two suggestions that could greatly improve the impact of the manuscript, but could also be considered as too much of extra-work.

1) Concerning Figure 3: The authors reports that expression of an activated forms of hayan or PSH can activate the Toll pathway through MP1. It would be great to know if this is the case in vivo in flies as the current knowledge suggests that only SPE is required. In short, do flies over-expressing activated forms of Hayan or PSH activate the Toll pathway? And if yes, it is really through MP1?

2) It would have been great to analyze the activation of the Toll pathway in both their model of sterile injury and upon septic injury by a gram-positive bacterium in Hayan, MP1 ans Psh, MP1 double mutants flies to better characterize the relation between these serine protases.

Minor recommendation

Line 81: add the fat body in addition to hemocytes and LG

Lines 88-93: maybe worth to mention the work on spz5

Line 157: The authors should provide a bit more information on their apoptosis deficient system.

Line 193: So far, Bomanins are not AMPs but rather host defense peptides (no detected microbicidal activity)

Line 203-204: the statement is too strong. Previous studies have shown that cleaved form of spz activates the Toll pathway (ex. Delotto and Delotto 1998). Just rephrase.

Line 241: Are the SP secreted in the extracellular space or do they activate spz in the secretory pathway (Golgi?) in cells? I would have expected that SP cleaves Spz extracellularly.

Line 288: for clarification, mention that those pro-forms are not active

Line 290: can the authors speculates why proHayan can cleave spz in S3B

Line 305 and earlier; it would have been great to know which SP genes are endogenously expressed in S2 cells.

Lines 369-370: it seems that the partially overlapping/redundant roles of Hayan and Psh has clearly been established by Dudzic 2019: Toll was not activated by proteases and septic injury in the double mutants flies.

Lines 374: Does over-expression of Hayan sufficient to activate the Toll pathway. Is it consistent with results published by Nam et al., EMBO ?. Here again, it would have great to know if over-expression of Hayan is sufficient to activate the Toll pathway in vivo in flies.

Line 292-396: The implication of lysosomes was not clear to me. Does the author mean that the activation of Psh by H2O2 take place inside a cell or is it linked to the release of proteases due to the breakage of lysosomes in the hemolymph that activated PSH extracellularly. Provide a bit more information.

Line 424; I would remove the term pathogen.

Reviewer #2: In this manuscript, Miura and coll. investigate activation of the Toll signaling pathway in drosophila in response to tissue damage. The manuscript is well written and I did not detect major problems in the experiments performed, or any major flaw in the interpretation of the results. Overall, the authors use a combination of genetics and biochemical experiments in S2 tissue culture cells to provide a refined model of the role of serine proteases acting upstream of the cytokine Spatzle (Spz) to activate Toll. This model emphasizes the role of Hayan and Persephone in a damage sensing pathway, in the absence of infection, and a more complex role than previously known for Melanization Protease 1 (MP1), which is able to cleave Spz in the S2 cell assay. While I enjoyed reading the manuscript, I could not help from feeling frustrated that the mechanism coupling ROS production to Psh activation is not addressed and one is left wondering about the possible involvement of the cathepsin 26-29p. Overall, I believe that this manuscript is a suitable candidate for publication in PLoS genetics provided the following points are addressed by the authors:

Major points

1) The role of the cathepsin 26-29p, which I believe is expressed in S2 cells, should be addressed.

2) Page 12, lane 249: the authors should justify why they do not consider a role for Easter

Minor points

3) Page 10, lane 204: the authors may want to tone done the claim that the importance of Spz cleavage has not been investigated in vivo (e.g., PMID: 10489372, 12872120).

4) Page 14, lane 290: please comment on the discrepancy between Fig5A,B (no cleavage of Pro-Spz in the presence of Pro-Psh) and FigS3B, and provide a more quantitative assessment than “strongly enhanced”.

**Have all data underlying the figures and results presented in the manuscript been provided?**

Reviewer #1: Yes

Reviewer #2: Yes

PLOS authors have the option to publish the peer review history of their article (what does this mean?). If published, this will include your full peer review and any attached files.

Reviewer #1: **Yes: **Bruno Lemaitre

Reviewer #2: No

---

## [Decision Letter · Decision Letter 1]

25 Apr 2023

Dear Dr Miura,

We are pleased to inform you that your manuscript entitled "Damage sensing mediated by serine proteases Hayan and Persephone for Toll pathway activation in apoptosis-deficient flies" has been editorially accepted for publication in PLOS Genetics. Congratulations!

Yours sincerely,

Norbert Perrimon

Academic Editor

PLOS Genetics

Gregory P. Copenhaver

Editor-in-Chief

PLOS Genetics

Comments from the reviewers (if applicable):

Reviewer's Responses to Questions

**Comments to the Authors:**

Reviewer #1: The paper has been improved and is now suitable for publication. This is an impressive piece of of work that sheds light on the serine protease cascade regulating the Toll pathway.

[I was just surprised that figure legends were in the results part....This does not help the reading.]

Reviewer #2: I thank the authors for taking the time to perform additional experiments to address the points I raised in my initial review.

**Have all data underlying the figures and results presented in the manuscript been provided?**

Reviewer #1: Yes

Reviewer #2: Yes

PLOS authors have the option to publish the peer review history of their article (what does this mean?). If published, this will include your full peer review and any attached files.

Reviewer #1: **Yes: **Bruno Lemaitre

Reviewer #2: No

**Data Deposition**

http://datadryad.org/submit?journalID=pgenetics&manu=PGENETICS-D-22-01327R1

**Press Queries**

---

## [Editor Report · Acceptance letter]

18 May 2023

PGENETICS-D-22-01327R1 

Damage sensing mediated by serine proteases Hayan and Persephone for Toll pathway activation in apoptosis-deficient flies 

Dear Dr Miura, 

We are pleased to inform you that your manuscript entitled "Damage sensing mediated by serine proteases Hayan and Persephone for Toll pathway activation in apoptosis-deficient flies" has been formally accepted for publication in PLOS Genetics! Your manuscript is now with our production department and you will be notified of the publication date in due course.

With kind regards,

Timea Kemeri-Szekernyes

PLOS Genetics

On behalf of:
